# Unveiling AI's Blind Spots:
# An Oracle for In-Domain, Out-of-Domain, and Adversarial Errors

**Shuangpeng Han** [1] [2]   **Mengmi Zhang** [1] [2]

## Abstract

AI models make mistakes when recognizing images—whether in-domain, out-of-domain, or adversarial. Predicting these errors is critical for improving system reliability, reducing costly mistakes, and enabling proactive corrections in real-world applications such as healthcare, finance, and autonomous systems. However, understanding what mistakes AI models make, why they occur, and how to predict them remains an open challenge. Here, we conduct comprehensive empirical evaluations using a "mentor" model—a deep neural network designed to predict another "mentee" model's errors. Our findings show that the mentor excels at learning from a mentee's mistakes on adversarial images with small perturbations and generalizes effectively to predict in-domain and out-of-domain errors of the mentee. Additionally, transformer-based mentor models excel at predicting errors across various mentee architectures. Subsequently, we draw insights from these observations and develop an "oracle" mentor model, dubbed SuperMentor, that can outperform baseline mentors in predicting errors across different error types from the ImageNet-1K dataset. Our framework paves the way for future research on anticipating and correcting AI model behaviors, ultimately increasing trust in AI systems. Our data and code are available at here.

## 1. Introduction

AI models are prone to making errors in image recognition tasks, whether dealing with in-domain, out-of-domain

(OOD), or adversarial examples. In-domain errors occur when models misclassify familiar data within the training domain, while OOD errors arise when faced with unseen or out-of-domain data. Adversarial errors are particularly concerning, as they result from carefully crafted perturbations designed to mislead the model.

Accurately predicting these errors is critical to enhancing the robustness and reliability of AI, especially in high-stakes real-world applications such as healthcare (Habehh & Gohel, 2021), finance (Mashrur et al., 2020), and autonomous driving (Huang et al., 2022). Proactively identifying potential errors enables more efficient corrections, reducing costly mistakes and safeguarding against catastrophic failures. By predicting when models are likely to err, we can implement strategies that either mitigate or entirely avoid the risks associated with those errors, ultimately leading to more trustworthy AI deployments.

Understanding the specific types of errors AI systems make, the reasons why they make these errors, and most importantly, how to predict these errors remains an unresolved challenge. Existing literature on error monitoring systems for AI models encompasses various approaches, including uncertainty estimation (Nado et al., 2021; Lakshminarayanan et al., 2017), anomaly detection (Bogdoll et al., 2022), outlier detection (Boukerche et al., 2020), and out-of-domain detection (Yang et al., 2024). While these methods are crucial for assessing model reliability, they mainly focus on determining whether a given data point falls outside the scope of the model's training. Thus, these approaches misalign with our primary objective of predicting whether AI models will make mistakes, as models can err on familiar data while behaving correctly on out-of-scope samples.

Subsequent research in out-of-domain detection has demonstrated that a model's accuracy is often correlated with how far the data deviates from in-domain samples (Hendrycks & Dietterich, 2019; Shankar et al., 2021; Li et al., 2017). These methods typically rely on predefined metrics, such as model parameter distances (Yu et al., 2022), model disagreements (Jiang et al., 2022; Madani et al., 2004) and confidence scores (Guillory et al., 2021), which limits their ability to generalize predictions

---

[1]Deep NeuroCognition Lab, College of Computing and Data Science, Nanyang Technological University, Singapore [2]Agency for Science, Technology and Research (A*STAR), Singapore. Correspondence to: Mengmi Zhang <mengmi.zhang@ntu.edu.sg>.

*Proceedings of the 42$^{nd}$ International Conference on Machine Learning*, Vancouver, Canada. PMLR 267, 2025. Copyright 2025 by the author(s).

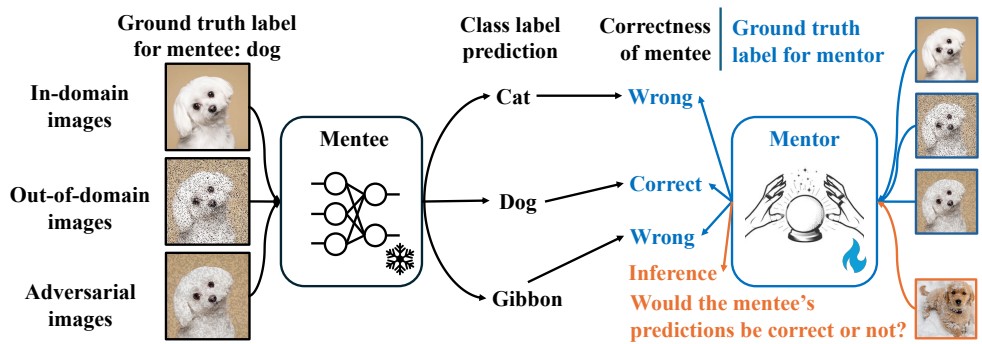

*Figure 1.* **AI models make mistakes and an "oracle" mentor model predicts when they will happen.** A "mentee" neural network (black) was trained for multi-class image recognition, but it can still misclassify in-domain, out-of-domain, and adversarial images. For instance, it might mislabel an in-domain dog image as a cat. The mentor model (blue), inputting the same images as the mentee, predicts whether the mentee will make a mistake. For example, if the mentee incorrectly labels an adversarial dog image, the mentor's ground truth label is "wrong"; conversely, if the mentee correctly labels an out-of-domain dog image, the mentor's label is "correct". The mentee's parameters are frozen (snowflake), while the mentor's are trainable (fire). During inference (orange), the mentor predicts whether the mentee will make an error on test images that have never been seen by both the mentee and the mentor.

across various data types, including errors arising from in-domain data or adversarial attacks (Szegedy et al., 2014). In parallel, earlier efforts in trustworthiness prediction mostly depend on shallow neural networks (Corbière et al., 2019; Qiu & Miikkulainen, 2022; Jiang et al., 2018) or carefully-designed loss functions (Luo et al., 2021). However, they do not investigate how different error types influence trustworthiness prediction.

Another line of research improves the robustness of the AI models with adversarial training approaches (Ilyas et al., 2019; Gowal et al., 2020; Balunović & Vechev, 2020); however, these approaches primarily focus on improving the model's overall performance rather than predicting when errors may occur in the models. Moreover, unlike selective prediction (Geifman & El-Yaniv, 2017), rejection learning (Cortes et al., 2016), and learning to defer (Madras et al., 2018), which jointly train the selection/rejection function with the mentee, our method trains the mentor and mentee independently. This separation is especially beneficial when mentee training is time- and resource-intensive or when its training data is inaccessible.

Different from all these works, we delve into the underlying principles of errors generated by AI models in the task of image classification with another AI model. Specifically, we designate the AI model that predicts errors as the **mentor** and the AI model being evaluated for performance as the **mentee**. The mentor strives to predict whether the mentee makes a mistake for any given data. See **Fig. 1** for the illustration of the problem setup. Training the mentor on the error patterns made by the mentee can potentially reveal the strengths and weaknesses of the mentee's learned representations across various visual contexts.

Our main contributions are highlighted below:

**1.** We conduct an in-depth analysis of how training mentors on each of three distinct error types specified by the mentees—In-Domain (ID) Errors, Out-of-Domain (OOD) Errors, and Adversarial Attack (AA) Errors—affect the performance of error predictions over three increasingly complex image datasets CIFAR-10 (Krizhevsky et al., 2009), CIFAR-100 (Krizhevsky et al., 2009) and ImageNet-1K (Deng et al., 2009). Our results reveal that training mentors with adversarial attack errors from the mentee has the most significant impact on improving the mentor's error prediction accuracy.

**2.** We investigate how various mentor model architectures affect error prediction performance. Our experiments demonstrate that transformer-based mentor models outperform other architectures in predicting errors.

**3.** We explore how varying levels of distortion in OOD and adversarial images affect the accuracy of error predictions. The findings indicate that training mentors on images with small perturbations can improve error prediction accuracy. In addition, we show that a mentor trained to learn error patterns from one mentee can successfully generalize its error predictions to another mentee.

**4.** Based on our findings from points 1 to 3, we present the SuperMentor model, which predicts errors across diverse mentee architectures and error types. Experimental results show that SuperMentor outperforms baseline mentors, demonstrating its superior error-predictive capabilities.

## 2. Related work

**Error monitoring systems for AI models.** With the growing deployment of AI models across diverse fields, ensuring their reliability and understanding their limitations has become increasingly crucial. This has led to numerous research in safe AI such as uncertainty estimation (Nado et al., 2021; Lakshminarayanan et al.,

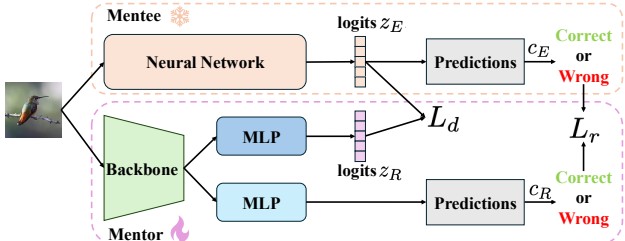

*Figure 2.* **Overview of a mentor model.** Given a fixed mentee model (snowflake), the mentor model takes an input image and uses a pre-trained backbone on ImageNet-1K (Deng et al., 2009) to extract features. The feature maps are then processed in two streams via multi-layer perceptrons (MLP)s. The output logits $z_R$ from one stream are compared with the mentee's output logits $z_E$ using a distillation loss $L_d$. The other stream performs a binary prediction of whether the mentee makes a mistake or not. The prediction is supervised by a logistic regression loss $L_r$. The parameters of MLPs in the two streams are not shared.

2017), anomaly detection (Bogdoll et al., 2022), outlier detection (Boukerche et al., 2020) and out-of-domain detection (Yang et al., 2024). Unlike these areas, which mainly aim to predict whether the input data falls outside the training domain, our focus is on monitoring and predicting errors in AI models by determining whether the model's output is correct, irrespective of whether the data comes from the training domain. To detect whether the input data is out of scope, the prior approaches mainly rely on softmax outputs (Granese et al., 2021; Hendrycks & Gimpel, 2017; DANG et al.), activations from network layers (Wang et al., 2020; Cheng et al., 2019; Ferreira et al., 2023), shallow neural networks (Corbière et al., 2019; Qiu & Miikkulainen, 2022; Jiang et al., 2018), and carefully-designed loss functions (Luo et al., 2021) in applications such as object detection (Kang et al., 2018) and trajectory prediction (Shao et al., 2023; 2024). However, these methods often rely on manually defined metrics to estimate the likelihood of a mentee making mistakes, or they fail to examine how different error types affect error prediction. In contrast, our strategy employs a separate deep neural network to automatically learn and approximate the mentee's decision boundaries for specific error types, providing an end-to-end trainable framework for error prediction. Moreover, our research direction differs from selective prediction (Geifman & El-Yaniv, 2017), rejection learning (Cortes et al., 2016), and learning to defer (Madras et al., 2018) by avoiding joint training with the mentee and requiring no knowledge of the mentee's training data. Besides, although rejecting a mentee's unreliable predictions and predicting the correctness of a mentee's outputs are often positively correlated, they remain distinct tasks.

**Out-of-domain detection.** Our research on predicting mentee errors is closely related to out-of-domain detection

in error monitoring systems, though it differs in several key aspects. As highlighted by (Guérin et al., 2023), error prediction is distinct from OOD detection (Liu et al., 2020a; Sun et al., 2021; Lee et al., 2018; Sun et al., 2022) in their objectives. While OOD detection aims to detect whether the given data comes from the same domain as the training set, the aim of error prediction is to learn whether the mentee will make a mistake on the given data. In other words, out-of-domain data may not necessarily cause the model to err, and model errors can also occur on in-domain data.

Recent studies (Hendrycks & Dietterich, 2019; Shankar et al., 2021; Li et al., 2017) have shown that a model's accuracy on a given dataset is often correlated with how far the data deviates from in-domain samples. However, these studies typically rely on pre-defined metrics, such as model parameter distances (Yu et al., 2022), model disagreements (Jiang et al., 2022; Madani et al., 2004), confidence scores (Guillory et al., 2021), domain-invariant representations (Chuang et al., 2020), and domain augmentation (Deng et al., 2021a), limiting their ability to generalize error prediction beyond in-domain data. In contrast, our mentor is capable of predicting both OOD and in-domain errors for a mentee. Additionally, our mentor is an AI model trained end-to-end without relying on manually defined criteria.

**Adversarial attack and defense.** In addition to OOD error, (Szegedy et al., 2014) discovered that deep neural networks can be fooled using input perturbations of extremely low magnitude. Building upon this finding, a substantial number of adversarial attacks have been proposed, including white-box attacks (Goodfellow et al., 2015; Mądry et al., 2017; Carlini & Wagner, 2017; Schwinn et al., 2023; Gao et al., 2020), black-box attacks (Uesato et al., 2018; Rahmati et al., 2020; Brendel et al., 2021; Chen et al., 2020), and backdoor attacks (Liu et al., 2020b; Xie et al., 2019; Kolouri et al., 2020). To defend against these adversarial attacks, various defence mechanisms (Qin et al., 2020; Deng et al., 2021b; Liu et al., 2019) have been developed to withstand or detect adversarial inputs. Furthermore, although the primary objective of adversarial attacks is to deceive AI models, there are instances where adversarial perturbations are exploited to enhance the model performance — a technique known as adversarial training (Ilyas et al., 2019; Gowal et al., 2020; Balunović & Vechev, 2020). Unlike adversarial training, which involves using adversarial samples to train the mentee, our approach focuses on teaching mentors to learn the mentee's error patterns revealed by these adversarial attack samples.

## 3. Experimental setups

We denote the mentor and mentee networks as $f_R(\cdot)$ and $f_E(\cdot)$ respectively. We also define $\mathcal{X}$ as the domain-specific

set containing all the test images for a mentee, and $\mathcal{Y}$ as their ground-truth object class labels. Therefore, a mentor is expected to make perfect predictions about the correctness of the mentee's responses (1 for "correct" and 0 for "wrong") given any image $x$ from $\mathcal{X}$:

$$\forall x \in \mathcal{X}, f_R(x) = \begin{cases} 1 & \text{if } f_E(x) = y, \\ 0 & \text{otherwise.} \end{cases} \quad (1)$$

where $y \in \mathcal{Y}$ is the ground-truth object class label of the corresponding image $x$.

### 3.1. Mentors

**Model architecture:** We propose mentor models, as illustrated in **Fig. 2**. Given an input image, the backbone of a mentor model extracts features from the input image. We adopt either of the two backbones for the feature extractors of mentors: a 2D Convolutional Neural Network (2D-CNN) ResNet50 (He et al., 2016) and a transformer-based ViT (Dosovitskiy et al., 2020). The extracted feature maps are further processed in two streams implemented as multi-layer perceptrons (MLP)s. The parameters of the MLPs in the two streams are not shared.

The first stream generates logits $z_R$ by predicting the probability distribution of a mentee over all the object classes when the mentee classifies the given image. The mentee network is kept fixed while training the mentor. Let us define the mentee's output logit as $z_E$. We introduce the distillation loss proposed by (Hinton et al., 2015): $L_d = T^2 \cdot KL(\sigma(\frac{z_E}{T})||\sigma(\frac{z_R}{T}))$ to align $z_R$ with $z_E$, where $KL(\cdot||\cdot)$ represents Kullback-Leibler divergence. $\sigma(\cdot)$ is the softmax function. $T = 1.0$ is the temperature, which controls the smoothness of the soft probability distribution.

In the second stream, the mentor is prompted to predict whether the mentee will make a mistake on the given image or not. We denote the predicted binary label as $c_R$, where 1 indicates that the mentee does not make a mistake and vice versa for 0. This prediction is supervised by $L_r = -[c_E \log(z_p) + (1 - c_E) \log(1 - z_p)]$, where $c_E$ is the ground truth correctness label of a mentee and $z_p$ represents the mentor's predicted probability that the mentee's prediction is correct. The overall loss is $L = \alpha L_r + (1 - \alpha)L_d$, where $\alpha = \left(\frac{n}{N}\right)^q$ controls the dynamic weighting between $L_d$ and $L_r$ over training epochs. Here, $n$ denotes the current training epoch, $N$ is the total number of training epochs, and $q$ governs the rate at which $\alpha$ evolves throughout training.

**Training and implementation details:** All mentors are trained on NVIDIA RTX A6000 GPUs, utilizing the AdamW optimizer (Loshchilov & Hutter, 2019) with a cosine annealing scheduler (Loshchilov & Hutter, 2022). The initial learning rate is set to $1 \times 10^{-4}$ for ResNet50

mentors and $3 \times 10^{-5}$ for ViT mentors. All mentors load the weights of the feature extractor pre-trained on the ImageNet-1K dataset for 1000-way image classification tasks (Deng et al., 2009) and further fine-tune on the error prediction task. During training, images are resized and center-cropped to $224 \times 224$ pixels. All mentors are trained for 30 epochs with a batch size of 32 on the CIFAR-10 and CIFAR-100 and 384 on the ImageNet-1K. The value of $q$ in the loss function is chosen for mentors to approach optimal performance. Specifically, $q$ is set to 1.0 for the mentors on CIFAR-10, 2.0 for those on CIFAR-100, 3.0 for ResNet50 mentors on ImageNet-1K (Deng et al., 2009), and 0.1 for ViT mentors on ImageNet-1K.

### 3.2. Mentees and their datasets

We employ two architectures as the mentees' backbones: ResNet50 (He et al., 2016), which is a 2D-Convolutional neural network (2D-CNN), and ViT (Dosovitskiy et al., 2020), which is a transformer based on self-attention mechanisms.

To train and test our mentees, we include three image datasets of varying sizes and follow their standard data splits: CIFAR-10 (C10, (Krizhevsky et al., 2009)) with 10 object classes, CIFAR-100 with 100 object classes (C100, (Krizhevsky et al., 2009)) and ImageNet-1K with 1000 object classes (IN, (Deng et al., 2009)). The multi-class recognition accuracy on the standard test sets of C10, C100 and IN datasets are 96.98%, 84.54%, 76.13% for the ResNet50 mentee and 97.45%, 86.51%, 81.07% for the ViT mentee respectively. The parameters of the mentees are frozen in all the experiments conducted on mentors.

### 3.3. Datasets for training and testing mentors

The mentor's objective is to predict whether the mentee will misclassify a given image, regardless of its source. The mentor is trained on correctly and wrongly classified images by a mentee. Next, we introduce how these images are curated and collected. A mentee may encounter various types of errors when dealing with real-world data. To explore which error types most effectively reveal the mentee's learning patterns, we categorize errors into three types: (1) errors from in-domain images, (2) errors from out-of-domain images, and (3) errors from adversarial images generated by adversarial attacks. Next, we introduce these three error types in detail.

**In-Domain (ID) Errors** occur on data that come from the same domain as the mentee's training dataset. Specifically, errors on images from the standard validation set of IN or the test sets of C10 and C100 are considered ID errors. Along with the correctly classified images from these standard test sets, we create three datasets for a mentor: **IN-ID**, **C10-ID**, and **C100-ID**, following the naming convention of

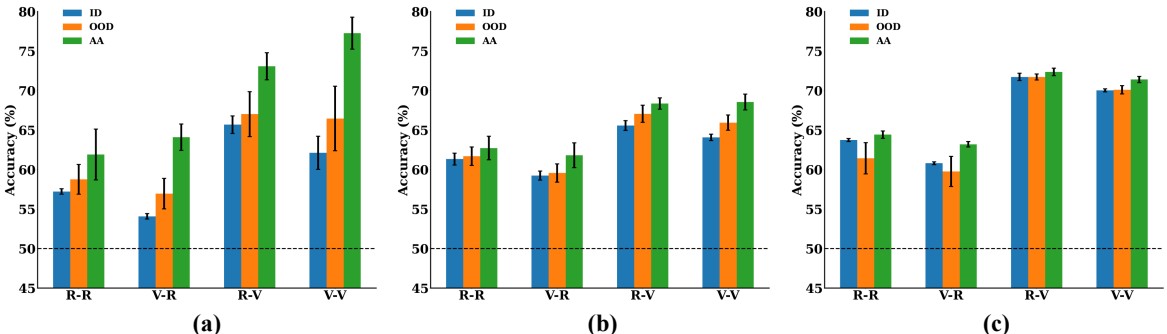

*Figure 3.* **Mentors trained on adversarial images of a mentee outperform mentors trained on OOD and ID images of the same mentee.** Average accuracy of a mentor trained on one type of error of a mentee for **(a)** C10, **(b)** C100 and **(c)** IN datasets is presented. Three types of errors made by a mentee are categorized based on in-domain (ID, blue), out-of-domain (OOD, orange), and images generated by adversarial attacks (AA, green). In each subplot, the labels on the x-axis are interpreted as [mentee]-[mentor], where "V" and "R" represent ViT and ResNet50 architectures for a mentee or a mentor respectively. Error bars indicate the standard deviation. The dotted black line indicates the chance level. See **Sec. 3.3** and **Sec. 3.4** for error types and the evaluation metric. The four sets of bars in each subfigure correspond to the confusion matrices shown in subfigures (a), (b), (c), and (d) of **Appendix, Fig. S4- S6**.

[Dataset]-[Error Type].

**Out-of-domain (OOD) Errors** refer to errors that arise when the mentee encounters data outside the training domain. To obtain OOD samples of a dataset, we adopt four types of image corruptions from (Hendrycks & Dietterich, 2019): **speckle noise (SpN)** (noise category), **Gaussian blur (GaB)** (blur category), **spatter (Spat)** (weather category), and **saturate (Sat)** (digital category). The noise levels can vary and we select level 1 for image corruptions as specified in (Hendrycks & Dietterich, 2019) by default. As noise levels increase, distortions on OOD images become more pronounced, causing the mentee to make more errors.

Following the naming conventions of [Dataset]-[Error Type]-[Error Source], we collect correctly and wrongly classified OOD samples based on C10 images of a mentee and curate four datasets for a mentor: **C10-OOD-SpN**, **C10-OOD-GaB**, **C10-OOD-Spat** and **C10-OOD-Sat**. Without the loss of generality, we can also curate four datasets each for a mentor based on C100 and IN images of a mentee.

**Adversarial Attack (AA) Errors.** Errors from adversarial images are specifically generated by adversarial attack methods to mislead or confuse the mentee. Given our assumption that the mentor has full access to the student model's parameters, we focus exclusively on white-box adversarial attacks as they typically produce more subtle yet effective perturbations compared to their black-box counterparts. To generate adversarial images, we employ four untargeted adversarial attack methods: **PGD** (Mądry et al., 2017) creates adversarial examples by repeatedly taking steps along the loss gradient; **CW** (Carlini & Wagner, 2017) attempts to minimize the $L_2$ norm of the perturbation while ensuring misclassification. **Jitter** (Schwinn et al., 2023) adds Gaussian noise to the output logits to encourage a diverse set of target classes for the attack. **PIFGSM** (Gao

et al., 2020) crafts patch-wise noise instead of pixel-wise noise. We set $c = 1.0$ in the CW attack, and perturbation bound $\epsilon = \frac{1}{255}$ for other attacks by default. See their papers for these hyper-parameter definitions. Intuitively, the attacks are stronger with higher hyper-parameter values; hence, the mentees make more mistakes.

Note that adversarial attacks are not always successful, and mentees can still correctly classify some adversarial images. We collect both the correctly and incorrectly classified adversarial images by a mentee based on C10 images, curating four datasets for the mentor: **C10-AA-PGD**, **C10-AA-CW**, **C10-AA-Jitter**, **C10-AA-PIFGSM**. Similarly, we can also curate four datasets each for a mentor based on C100 and IN images of a mentee.

**Training and test splits for mentors.** Given the three types of errors described above, we partition the data for mentors into training and testing sets using a 70/30 split. For example, 70% of the original C10 evaluation images, along with their corresponding error-modified versions based on mentee performance, are used for training, while the remaining 30% are reserved for testing. This ensures that mentor models are trained and evaluated on samples derived from the same set of original images, without any overlap in the original image content, but with distinct domain shifts. The same strategy is consistently applied to the C100 and IN datasets. To mitigate the effects of long-tailed class distributions, each training batch includes an equal number of correctly and incorrectly classified samples.

### 3.4. Baselines and evaluation metric

**Baselines.** We include seven baseline methods for error prediction of a mentee. 1) **Self Error Rate (SER)** predicts the correctness of the mentee's outputs by randomly assigning "correct" or "wrong" based on the mentee's in-domain accuracy; 2) **Maximum Class**

**Probability (MCP)** (**Hendrycks & Gimpel, 2017**) classify the mentee's predictions as correct if its MCP exceeds a predefined threshold. 3) **Class Probability Entropy (CPE)** evaluates a mentee's prediction as correct if its entropy of the probability distribution over all classes is below a predefined threshold. 4) **Distance To Centroid (DTC)** considers a prediction correct if its feature embedding is within a predefined distance from the class centroid. 5) **ConfidNet** (**Corbière et al., 2019**) predicts failure by learning True Class Probability (TCP) on the training samples. 6) **TrustScore** (**Jiang et al., 2018**) employs Trust Score to quantify the agreement between the classifier and a modified nearest-neighbor classifier.7) **Steep Slope Loss (SSL)** (**Luo et al., 2021**) develops a neural network for predicting trustworthiness by utilizing a steep slope loss function. See **Appendix, Sec. A** for more details.

**Evaluation metric.** To assess the performance of mentors, we report their error prediction accuracy on the test set corresponding to each specified error source. For instance, a mentor trained on the C10-ID training set is evaluated on the C10-OOD-SpN test set. The error prediction accuracy is calculated by averaging the mentor's accuracies on the samples that the mentee correctly classified and those that the mentee incorrectly classified. However, since a mentee can make mistakes across various real-world scenarios, a mentor must accurately predict errors across all error types. Therefore, we compute the average accuracy, named as *Accuracy (%)* of a mentor across all test sets, including one ID error, four OOD errors, and four AA errors. See **Appendix, Sec. B** for an example calculation of the average accuracy of a mentor.

## 4. Results

### 4.1. Training on specific errors of mentees impacts the performance of mentors

A mentee's mistakes can reveal their learning biases, behaviors, or traits. Here, we investigate which types of errors offer the most insight into understanding a mentee's decision boundaries during image recognition tasks. We train mentors with identical architectures on datasets containing specific error types made by the mentee across C10 (**Fig. 3(a)**), C100 (**Fig. 3(b)**), and IN (**Fig. 3(c)**). For instance, if a mentor trained on C10-OOD achieves higher accuracy in error prediction compared to one trained on C10-ID, this suggests that in-domain errors provide less diagnostic information about the mentee's decision-making process than out-of-domain errors. Both mentors and mentees may have the same or different backbones, such as ResNet50 (R) or ViT (V).

As shown in **Fig. 3**, over C10, C100, and IN images, the high average accuracies for mentors trained on adversarial attack (AA) errors indicates that these AA errors offer

deeper insights into the mentee's decision process compared to out-of-domain (OOD) and in-domain (ID) errors. To validate this, we first conducted an ANOVA test on the three groups of results (ID, OOD, and AA) for each of the three datasets (C10, C100, and IN). In all cases, the p-values are below 0.05, indicating a statistically significant difference in error prediction performance among mentors trained on different error types. Next, we performed two-tailed t-tests to compare AA-trained mentors against ID-trained and OOD-trained mentors across four [mentee]-[mentor] configurations: R–R, V–R, R–V, and V–V. All pairwise p-values are below 0.05, confirming that AA-trained mentors significantly outperform those trained on ID or OOD error types. In addition, it is worth noting that, in some cases, mentors trained on OOD errors slightly outperformed those trained on ID errors, although both still performed worse than those trained on AA errors. Interestingly, our experiments reveal that training mentors exclusively on AA errors performs marginally worse than joint training on all three error types (see **Appendix, Sec. C**). This result underscores a significant overlap of mentee error patterns among ID, OOD, and AA data.

**Loss landscape analysis.** A loss landscape of a mentee reflects how a mentee's loss function behaves across different parameter configurations. Mentors' performance offers insights into the structure of a mentee's loss landscape. Consistent with (**Ilyas et al., 2019**), the high accuracy of mentors trained on AA errors suggests that adversarial images lie closer to the mentee's decision boundary, enabling more accurate prediction of the mentee's mistakes and a deeper understanding of the loss landscape. Similarly, OOD data aids mentors in learning boundaries by shifting ID samples closer to the boundary. However, it does not explore the boundary as thoroughly as adversarial images. ID data, with fewer samples near the boundary, provides more limited exploration compared to adversarial images. To support our claims, we include quantitative loss landscape analyses in **Appendix, Sec. D**. The results show that mentors trained on AA errors exhibit a wider loss landscape compared to those trained on ID or OOD errors.

### 4.2. Mentor architectures matter in error predictions

To computationally model the decision boundary of a mentee using a mentor, the mentor requires more complex architectures with a larger number of parameters than the mentee. Indeed, from **Fig. 3**, over all the datasets, we observed that utilizing ViT (V) as the mentor backbone consistently achieves higher accuracy across all error types of ViT-based and ResNet-based mentees compared to the mentor based on ResNet50 (R). One example of this performance disparity is observed in the context of the adversarial attack error type on C10. The mentors leveraging a ViT backbone achieves 73.1% accuracy in

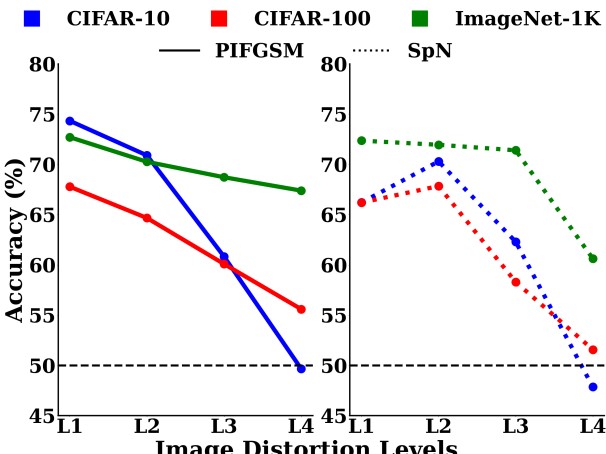

*Figure 4.* **A mentor's accuracy is heavily influenced by the levels of image distortions introduced by out-of-domain perturbations and adversarial attacks.** ViT mentor's accuracy is a function of varying image distortion levels from PIFGSM (Gao et al., 2020) and Speckle Noise (SpN) (Hendrycks & Dietterich, 2019) to the C10, C100 and IN images of a ResNet50-based mentee. The black dashed line indicates the chance level.

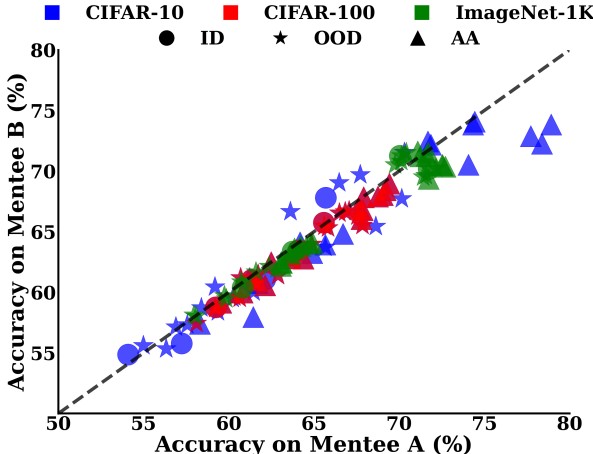

*Figure 5.* **Mentors can generalize their error predictions across different mentee architectures.** Mentors trained on mentee A's predictions (x-axis) are evaluated against the predictions from mentee B (y-axis). Each marker is a generalization experiment of a mentor trained on different error types (marker shapes) in different image datasets (colours) of a mentee. The black dash line indicates the diagonal.

predicting the correctness of ResNet50 mentee's outputs, which is significantly higher than the 61.9% accuracy of the ResNet-based mentor (compare R-R versus R-V).

**Loss landscape analysis.** The performance difference between mentors' architectures is due to ViT's superior ability to identify features from error patterns. Its self-attention mechanism captures complex relationships among data samples, providing a deeper understanding of the mentee's loss landscape, particularly in modelling irregular, rugged landscapes with sharp peaks and valleys. To validate this point, we include loss landscape analyses in **Appendix, Sec. D**. Results show that ViT mentors possess a notably broader loss landscape than ResNet50 mentors.

### 4.3. Training on images with smaller perturbations helps error predictions

Although adversarial images have been demonstrated to aid in error prediction (**Sec. 4.1**), it remains unclear whether adversarial images with varying degrees of image distortion exhibit the same effect. A straightforward method to regulate the level of image distortion caused by adversarial attacks is to set the perturbation bound $\epsilon$. We employ four corruption levels by setting $\epsilon = \frac{1}{255}, \frac{2}{255}, \frac{4}{255}$, and $\frac{8}{255}$. We use the adversarial attack PIFGSM as an example since the error patterns from PIFGSM are most effective for the mentor's prediction (see **Appendix, Fig. S4- S6**). As shown in **Fig. 4**, the mentor's accuracy significantly decreases as the distortion level increases. In particular, for the C10-AA-PIFGSM, the accuracy at level 1 is 74.3%, which is notably higher than 49.7% at level 4. Our findings suggest that adversarial attacks employing

smaller perturbations yield more benefits for mentor error prediction. This phenomenon can be attributed to the fact that adversarial images with minimal perturbations maintain closer proximity to the decision boundary of a mentee.

Building on the findings above, we investigate whether the mentor's performance is influenced by how far OOD images are from the ID data. Specifically, we aim to determine whether the degree of deviation from the training domain impacts the mentor in a similar way to our observations on adversarial images. To explore this, we analyze images corrupted with Speckle Noise (SpN) and adjust the standard deviation $\sigma$ of SpN to 0.01, 0.06, 0.15, and 0.6, representing four distinct levels of distortion. The outcomes are depicted in **Fig. 4**. We observe that the mentor's accuracy improves as the distortion introduced by SpN decreases. For example, the mentor achieves an accuracy of 66.2% on level 1 of C10-OOD-SpN, while the accuracy drops significantly to 47.9% on level 4 of C100-OOD-SpN. This suggests that OOD error types with smaller perturbations enhance the mentor's performance. However, unlike adversarial attacks, caution is necessary because the mentor's accuracy can plateau with extremely small distortion levels, as shown by the minimal difference in accuracy between levels 1 and 2 of SpN in **Fig. 4**.

### 4.4. Mentors generalize across mentees

In **Sec. 4.1**, mentors have demonstrated their ability to learn the error patterns of mentees. This observation raises an important question: can the error patterns learned from one mentee (mentee A) be generalized to another mentee (mentee B) when the two mentees employ different model

Figure 6. **Mentor analysis comprising: (a) performance comparison with baselines, (b) ablation of loss components, and (c) EigenCAM (Gildenblat & contributors, 2021; Muhammad & Yeasin, 2020) visualizations.** (a) compares the average accuracy of baselines and our SuperMentor on various error sources and severity levels from the IN dataset. The ResNet50 is used as the mentee, with mentors predicting its correctness. Results are reported as average accuracy with variance over 3 runs; best results are in bold. Full baseline results across hyperparameter settings are in **Appendix, Tab. S1**. Comparison with a ViT mentee is provided in **Appendix, Tab. S2**. (b) presents an ablation study where $L_d$ is the distillation loss (see **Sec. 3.1**) and $L_a$ is the alignment loss between mentor and mentee predictions. The ResNet50-based mentors are evaluated on ViT mentees, with each cell showing mean accuracy and standard deviation over 3 runs. Grey cells indicate our mentors' performance. (c) shows EigenCAM visualizations of the SuperMentor and ViT mentee on sample images from the brambling and baseball classes across IN-ID, IN-OOD-Spat, and IN-AA-PIFGSM datasets. For each sample, rows display the input image (Input), mentee's attention map (Mentee), and mentor's attention map (Mentor), using the first LayerNorm in the final transformer block as the visualization layer. Column headers indicate Ground Truth Label and Mentee's Prediction.

architectures? To explore this, we evaluate all 324 mentors on the alternate mentee. Specifically, mentors trained on the errors of the ResNet50 mentee are tested on the predictions of the ViT mentee, and vice versa. The outcomes of these evaluations are illustrated in **Fig. 5**. Surprisingly, most points lie near the dashed diagonal line, implying that the mentors' performance does not significantly deteriorate when evaluated on the predictions of different mentee architectures. This finding indicates that ResNet50 and ViT mentees tend to produce similar error patterns when trained on the same dataset.

In addition, we evaluate the mentors' generalization ability on natural adversarial samples introduced in (Hendrycks et al., 2021b), where each image is misclassified by multiple mentees simultaneously. See details in **Appendix, Sec. F**. Results show that, beyond natural adversarial examples, the mentors also exhibit robust generalization on non-natural adversarial examples.

### 4.5. Analysis on our SuperMentor reveals key insights

By drawing insights from observations in the subsections above, we propose an "oracle" mentor model, dubbed SuperMentor. We introduce the technical novelties of our SuperMentor below. First, as demonstrated in **Sec. 4.1** and **Sec. 4.3**, mentors trained on adversarial images with small perturbations of a mentee outperform those trained on OOD and ID images; thus, our SuperMentor adopts the training data from the PIFGSM error source of mentees with $\epsilon = \frac{1}{255}$. Second, since ViT has been proven to be a more effective architecture for mentors than ResNet50 (**Sec. 4.2**), SuperMentor adopts ViT as the backbone architecture.

We demonstrate the effectiveness of SuperMentor by

comparing it with the baselines introduced in **Sec. 3.4**. To conduct comprehensive evaluations, we extend image perturbations from different error sources to multiple severity levels. Specifically, we use corruption levels 1, 3, and 5 for SpN, GaB, Spat, and Sat error sources as specified in (Hendrycks & Dietterich, 2019); set $\epsilon$ to $\frac{1}{255}$, $\frac{4}{255}$, and $\frac{8}{255}$ for PIFGSM, Jitter, and PGD error sources; and vary the learning rate as 0.01, 0.1, and 1.0 for the CW error source.

As shown in **Fig. 6(a)**, our SuperMentor can achieve higher average accuracies across all severity levels and error types than baselines. Notably, as indicated in **Appendix, Tab. S1**, in the AA scenarios, SuperMentor has significantly higher accuracy than the baselines. For example, SuperMentor achieves an error prediction accuracy of 73.1% on IN-AA-PGD of the ResNet50 mentee, whereas the best baseline only reaches 64.7%. Baseline methods like SER, MCP, CPE, and DTC rely on fixed thresholds or predetermined values of manually defined criteria such as confidence or entropy, making them less adaptable to AA scenarios. In contrast, SuperMentor leverages deep neural networks to capture the complexity of error prediction. Additionally, methods like ConfidNet, TrustScore, and SSL are trained on ID error types. As shown in **Fig. 6(a)**, these baselines are inferior to SuperMentor, which is trained exclusively on AA data. This suggests that the source of error data used to train the mentor plays a more critical role than the choice of loss functions or the sophisticated training strategies in these baselines. Moreover, we provide the visualization of SuperMentor's embeddings based on the correctness of a mentee's prediction across three error types (**Appendix, Sec. G**). Two distinct clusters are observed between samples wrongly or correctly classified by the

mentee. This suggests that our SuperMentor is capable of accurately predicting the errors of a mentee regardless of the error sources.

Next, we examine the effect of the distillation loss $L_d$ (**Fig. 2**) on the mentors' performance. The results are presented in **Fig. 6(b)**. It is clear that excluding $L_d$ results in a decrease in mentors' accuracy across all datasets. For example, in the C10 dataset, the average accuracy of mentors decreases from $64.9\%$ to $59.3\%$. This suggests that $L_d$ encourages mentors to learn the fine-grained decision boundaries among different object classes of a mentee. Alternatively, instead of utilizing the mentee's logits, mentors can incorporate a cross-entropy loss to align the mentor's predicted object class labels with those of the mentee, denoted as $L_a$. From **Fig. 6(b)**, we observe that replacing $L_d$ with $L_a$ leads to a drop in accuracy. This is due to the fact that the mentee's logits contain more information than its class labels.

Finally, we provide EigenCAM (Gildenblat & contributors, 2021; Muhammad & Yeasin, 2020) visualization in **Fig. 6(c)** to illustrate the following two points: 1) The vulnerabilities of the mentee do not overlap with those of the mentors. As illustrated in **Fig. 6(c)**, the mentor does not merely replicate the mentee's learning patterns, as their activation maps for identical input images can differ significantly. For example, in the brambling images shown in **Fig. 6(c)**, the activation maps of the mentor and mentee diverge greatly when the input image is ID or subjected to AA using PIFGSM method. In the IN-AA-PIFGSM image, the mentee demonstrates vulnerability under adversarial attacks, whereas our SuperMentor does not exhibit such vulnerabilities on this AA image, consistently focusing on the brambling object. This indicates the vulnerabilities of mentors and mentees are distinct due to their non-overlapping objectives. 2) Our AI mentor can serve as a valuable diagnostic tool for AI mentees. As supported by the baseball images in **Fig. 6(c)**, although the mentee correctly classifies the ID image, its activation map does not concentrate on the baseball but instead highlights background cues (spurious features). When the input image is subjected to IN-AA-PIFGSM attacks, the mentee fails to make accurate predictions. In contrast, our SuperMentor effectively focuses on the stitches of the baseball regardless of whether the image is corrupted or adversarially attacked. This suggests that the mentor can identify vulnerabilities in the mentee, even though they both share the ViT architecture.

## 5. Discussion and Conclusion

In our work, we tackle the challenge of predicting errors of AI models through extensive empirical evaluations using an end-to-end trainable "mentor" model. This mentor is designed to assess the correctness of a

mentee's predictions across three distinct error types: in-domain errors, out-of-domain errors, and adversarial attack errors. Our results show that the mentor excels at learning from a mentee's errors on adversarial images with minimal perturbations and, surprisingly, generalizes well to both in-domain and out-of-domain predictions of the same mentee. Additionally, we highlight the effectiveness of transformer-based mentor architectures compared to 2D-CNN-based ones, demonstrating their superior generalization capabilities across mentees with diverse backbones. Lastly, we introduce the SuperMentor, which outperforms all existing mentor baselines.

Despite the promising results, our framework may raise concerns regarding the vulnerabilities of AI mentors. Indeed, vulnerability is a well-known challenge for all deep neural networks (DNNs), yet this has not hindered their transformative applications in the real world. For example, DNNs are used to detect diabetic retinopathy (Jumper et al., 2021; Alyoubi et al., 2020), outperforming traditional diagnostic methods. Similarly, our work uses one AI to diagnose the other AI, which is a valid and impactful use case. To demonstrate its real-world utility, we extended our experiments to a medical image classification task for colorectal cancer diagnosis (**Appendix, Sec. H**). Our mentor achieves high error prediction accuracy compared to all competitive baselines, showcasing its practical uses in high-stake applications. More importantly, beyond the OOD domains evaluated in **Sec. 4.5**, results in **Appendix, Sec. I** show that our SuperMentor remains robust to mentee error patterns across an extra set of five unseen OOD domains. Moreover, the vulnerabilities of the mentor AI do not overlap with those of the mentees since mentors operate independently of the mentee's architecture or training details, reducing the risk of shared weaknesses (see **Sec.4.5**).

Our work paves the way for several promising research directions in safe and trustworthy AI. First, while our current research focuses on image classification, there is potential to extend this approach to other vision and language tasks, such as object detection and machine translation. Second, future research could explore mutual learning between mentors and mentees, where mentors not only learn from the mentee's error patterns but also provide valuable feedback to help refine the mentee. Third, we can establish more rigorous evaluation criteria for mentors, broadening their predictive capabilities. For example, beyond predicting whether a mentee is likely to make errors, mentors could also forecast the specific types of errors a mentee may encounter. Fourth, drawing parallels with AI mentors for AI mentees, we can explore the possibility of using AI mentors to investigate recognition errors in humans and primates. Such studies could provide insights into error pattern alignment between biological and artificial intelligent systems.

## Acknowledgements

This research is supported by the National Research Foundation, Singapore under its NRFF award NRF-NRFF15-2023-0001 and Mengmi Zhang's Startup Grant from Nanyang Technological University.

## Impact Statement

As AI integrates into our daily lives and critical applications like healthcare, finance, and autonomous driving, ensuring its safety and reliability is imperative. Our research highlights the error patterns and characteristics of AI models for object recognition, paving the way for building trustworthy AI. By enabling one AI model to anticipate and correct another's behavior, our framework helps ensure AI systems operate in a more predictable and reliable manner. Overall, our work lays the foundation for developing systems capable of anticipating the errors of others, offering practical value in high-stakes real-world applications.

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

## A. Details of baselines

For performance comparison, we adopt seven baselines, as introduced in **Sec. 3.4**. Their details are outlined below.

**Self Error Rate (SER):** We predict the correctness of the mentee's outputs by referencing its accuracy on in-domain samples. For instance, if the mentee's in-domain accuracy is 70%, we use a random binary generator that produces 1 ("correct") with 70% probability and 0 ("wrong") with 30% probability, applying this to all testing scenarios.

**Maximum Class Probability (MCP) (Hendrycks & Gimpel, 2017):** As mentioned in **Sec. 3.1**, the mentee's output logit is denoted as $z_E$. The maximum class probability of the mentee's output is defined as $MCP(z_E) = \max \sigma(z_E))$ where $\sigma(\cdot)$ is the softmax function. The MCP indicates how confident the mentee is in its most probable prediction. We set a threshold $\gamma$ to distinguish between the mentee's confident and unconfident predictions. Specifically, if $MCP(z_E) > \gamma$, the mentee's prediction will be considered a correct prediction due to its high confidence; otherwise, it will be regarded as an incorrect prediction. We employ three predefined thresholds for $\gamma$: 0.5, 0.7, and 0.9.

**Class Probability Entropy (CPE):** The class probability entropy of the mentee's output is defined as $CPE(z_E) = H(\sigma(z_E))$, where $\sigma(\cdot)$ denotes the softmax function and $H(\cdot)$ represents the entropy measure quantifying the uncertainty in the probability distribution. A high entropy value signifies a high level of uncertainty in the mentee's predictions. The entropy reaches its maximum value, $MaxCPE(z_E)$, when the mentee's class probabilities in $\sigma(z_E)$ are equal. We define the uncertainty threshold as $\alpha \cdot MaxCPE(z_E)$, where $\alpha \in [0,1]$. If $CPE(z_E) < \alpha \cdot MaxCPE(z_E)$, the mentee's prediction is considered correct, indicating sufficient certainty in its prediction. Otherwise, the prediction is regarded as incorrect. We set three predefined values for $\alpha$: 0.01, 0.1 and 0.3.

**Distance to centroid (DTC):** The embedding generated by the mentee before the final binary classification layer represents the mentee's feature interpretation of each sample. To determine the feature centroid for each class, we first average the features of all testing samples within that class based on the mentee's predictions. Next, we calculate the L2 distance between each sample's feature and its corresponding class centroid, denoted as $d_s$. We establish a distance threshold $d$; if $d_s < d$, the mentee's prediction is considered correct since the sample is close to the class centroid. Otherwise, the prediction is deemed incorrect. We have set three predefined values for $d$: 10, 20, and 30.

**ConfidNet (Corbière et al., 2019):** A shallow failure prediction neural network that learns True Class Probability (TCP) from the training set instead of relying on Maximum Class Probability (MCP).

**TrustScore (Jiang et al., 2018):** It introduces a new score called trust score by measuring the agreement between the classifier and a modified nearest-neighbor classifier on the testing samples for error prediction.

**Steep Slope Loss (SSL) (Luo et al., 2021):** Training an AI model for trustworthiness prediction by leveraging a carefully-designed steep slope loss function.

## B. Example of calculating a mentor's average accuracy using the proposed evaluation metric

As an example of computing the accuracy of a mentor evaluated using the metric presented in **Sec. 3.4**, we consider a mentor trained on C10-AA-PIFGSM and evaluate it on all the following datasets with their accuracies of 10% on C10-ID, 20% on C10-OOD-SpN, 30% on C10-OOD-GaB, 40% on C10-OOD-Spat, 50% on C10-OOD-Sat, 60% on C10-AA-Jitter, 70% on C10-AA-PGD, 80% on C10-AA-CW, and 90% on C10-AA-PIFGSM testing samples, the average accuracy of this mentor is calculated as (10% + 20% + 30% + 40% + 50% + 60% + 70% + 80% + 90%) / 9 = 50%. For simplicity, we refer to this average accuracy across all nine error sources as **Accuracy**. A mentor randomly guessing whether a mentee's image classification is correct or incorrect for a given image would achieve an accuracy of 50%.

## C. Joint training

As mentioned in **Sec. 4.1**, we add an experiment by training a Vision Transformer (ViT)-based mentor model on the correctness of a mixture of ID, OOD, and AA data for the ResNet-based mentee model from the C10 dataset. The OOD data was corrupted with speckle noise (SpN), and the AA data was generated using PIFGSM. Compared to training the mentor solely on PIFGSM data which achieved an average accuracy of 74.3%, including ID and SpN data improved the average accuracy slightly to 77.5%. However, this joint training setup involves 21,000 samples—three times more than the 7,000 used for PIFGSM-only training. The marginal improvement observed in joint training may be attributed to the larger training sample size. This suggests that the quantity of samples is not the key factor in enhancing a mentor's error prediction

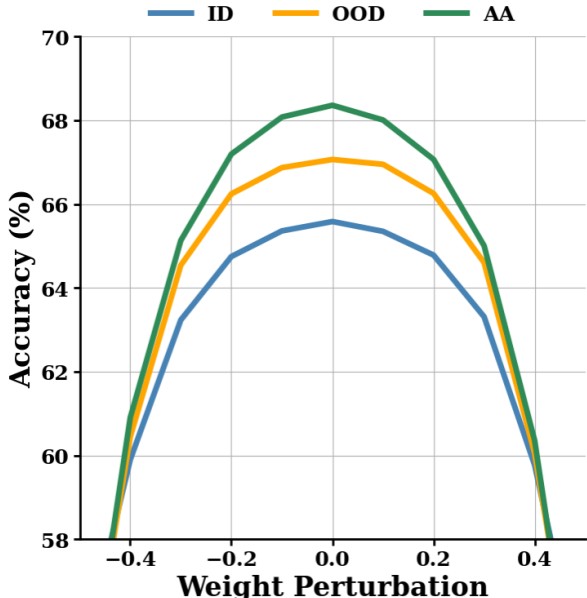

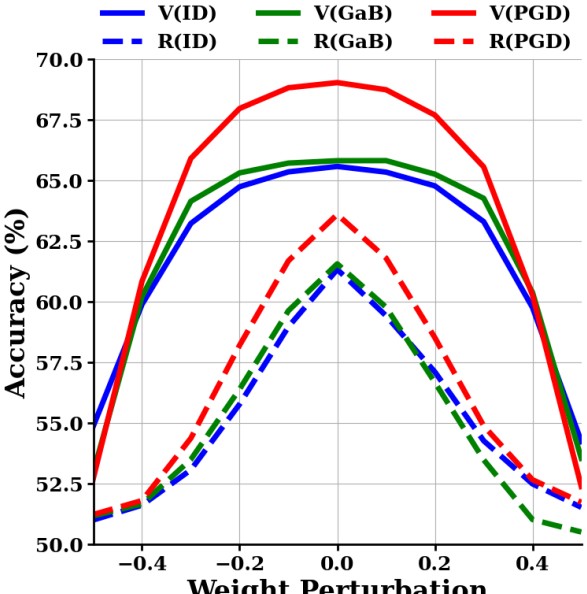

*Figure S1.* **Loss landscape analysis of mentors trained on three error types: in-domain (ID, blue), out-of-domain (OOD, orange), and adversarial attack images (AA, green) from the C100 dataset.** The average accuracies of mentors evaluated across various error sources under different weight perturbations are reported. The ResNet50 model serves as the mentee, and the ViT model is adopted as the mentor architecture.

*Figure S2.* **Loss landscape analysis of mentors with different backbones.** Mentors can be built with two different backbones: ViT (V, solid line) and ResNet50 (R, dotted line). We evaluate mentors trained on three error sources: C100-ID (blue), C100-OOD-GaB (green), and C100-AA-PGD (red). The legend follows the format [mentor backbone] ([training error source]). The average accuracies of mentors evaluated across various error sources under different weight perturbations are reported. The ResNet50 model serves as the mentee.

performance. Training mentors on samples that accurately capture the error patterns of mentees is crucial.

In addition, we introduce another mentor that is jointly trained on all error sources (ID, OOD, and AA, a total of 9 error sources) from the C10 dataset, achieving an average evaluation accuracy of 78.4%. In this scenario, all testing samples for the mentor are in distribution because the mentor has seen samples from every error source during training. Despite this, the improvement compared to a mentor trained solely on the PIFGSM error source (78.4% vs. 74.3%) is marginal. Consequently, this finding further indicates: (1) Training mentors exclusively on the AA error source offers a more data-efficient strategy for real-world applications. (2) Remarkably, a mentor trained solely on AA generalizes well to unseen error sources, achieving performance comparable to the upper bound obtained by jointly training on all error sources.

## D. Loss landscape analysis

We conducted two quantitative loss landscape analyses using the method from (Li et al., 2018) to support the claims in **Sec. 4.1** and **Sec. 4.2**. In these analyses, we apply small perturbations to the mentor's weights and observe the resulting changes in the mentors' average accuracy.

As shown in **Appendix, Fig. S1**, we examine mentors trained on three different error types: ID, OOD, and AA from the C100 dataset. The results indicate that mentors trained on AA errors exhibit a wider loss landscape than those trained on ID or OOD errors. This suggests that mentors trained with adversarial images capture the generic features for predicting the mentee's decision-making. This finding strongly supports our loss landscape analysis discussed in **Sec. 4.1**.

The second loss landscape analysis in **Appendix, Fig. S2** examines the mentor's error prediction performance using two different backbones: ResNet50 and ViT. The figure shows that mentors with ViT architectures have a considerably wider loss landscape compared to those with ResNet50. This further reinforces our claims in **Sec. 4.2** that transformer-based mentor models outperform their ResNet-based counterparts in error prediction.

# E. More comparisons between SuperMentor and baselines

As noted in the caption of **Fig. 6(a)**, baseline performances across multiple hyperparameter configurations are indicated in **Appendix, Tab. S1**. In addition, when ViT serves as the mentee, the SuperMentor's performance is shown in **Appendix, Tab. S2**. It can be seen that our SuperMentor can still achieve the best performance when the mentee has a backbone of ViT. This emphasizes the importance of the error source used during training.

*Table S1.* **Performance comparison of baselines (multiple hyperparameter configurations) and our SuperMentor on various error sources from the IN dataset.** The ResNet50 model serves as the mentee, with its correctness of predictions evaluated by mentors. Best results are in bold.

| | ID | SpN | GaB | Spat | Sat | PGD | CW | Jitter | PIFGSM | Average |
|---|---|---|---|---|---|---|---|---|---|---|
| SER | 50.3 | 49.9 | 50.1 | 50.2 | 50.2 | 50.1 | 50.1 | 50.1 | 50.3 | 50.1 |
| MCP ($\gamma = 0.5$) | 69.4 | **73.0** | **71.7** | 71.2 | 72.6 | 51.7 | 67.5 | 52.5 | 53.6 | 64.4 |
| MCP ($\gamma = 0.7$) | 77.5 | 71.6 | 70.7 | **73.9** | **76.5** | 50.3 | 70.0 | 50.4 | 53.6 | 65.1 |
| MCP ($\gamma = 0.9$) | **78.7** | 66.2 | 65.4 | 70.4 | 73.7 | 45.7 | 67.1 | 47.8 | 51.9 | 61.7 |
| CPE ($\alpha = 0.01$) | 69.3 | 57.0 | 56.9 | 60.3 | 62.7 | 43.4 | 58.5 | 47.0 | 48.9 | 54.9 |
| CPE ($\alpha = 0.1$) | 78.1 | 67.9 | 66.9 | 72.4 | 75.2 | 46.9 | **68.6** | 49.8 | 51.7 | 63.0 |
| CPE ($\alpha = 0.3$) | 63.8 | 71.9 | 69.6 | 69.2 | 68.9 | 51.1 | 65.1 | 52.4 | 52.3 | 62.6 |
| DTC ($d = 10$) | 52.6 | 52.4 | 51.9 | 52.6 | 52.4 | 52.3 | 51.1 | 50.5 | 51.3 | 51.8 |
| DTC ($d = 20$) | 51.6 | 50.2 | 50.1 | 50.7 | 50.9 | 51.0 | 50.0 | 50.1 | 52.8 | 50.8 |
| DTC ($d = 30$) | 50.0 | 50.0 | 50.0 | 50.0 | 50.0 | 50.0 | 50.0 | 50.0 | 50.0 | 50.0 |
| ConfidNet (Corbière et al., 2019) | 67.8 | 59.0 | 61.1 | 61.8 | 64.1 | 51.8 | 61.9 | 57.5 | 54.3 | 59.3 |
| TrustScore (Jiang et al., 2018) | 71.2 | 58.6 | 60.1 | 63.4 | 65.7 | 58.2 | 63.3 | 59.4 | 60.8 | 61.6 |
| SSL (Luo et al., 2021) | 68.6 | 66.2 | 69.2 | 69.6 | 68.4 | 64.7 | 63.4 | 63.2 | 67.9 | 66.7 |
| SuperMentor (ours) | 73.3 | 69.2 | 65.6 | 71.1 | 69.9 | **73.1** | 67.4 | **70.3** | **75.4** | **70.4** |

*Table S2.* **Performance comparison of baselines (multiple hyperparameter configurations) and our SuperMentor on various error sources from the IN dataset.** The ViT model serves as the mentee, with its correctness of predictions evaluated by mentors. Best results are in bold.

| | ID | SpN | GaB | Spat | Sat | PGD | CW | Jitter | PIFGSM | Average |
|---|---|---|---|---|---|---|---|---|---|---|
| SER | 49.8 | 50.2 | 49.9 | 50.1 | 50.0 | 49.7 | 50.1 | 50.1 | 49.9 | 50.0 |
| MCP ($\gamma = 0.5$) | 70.0 | 75.0 | 73.1 | 73.5 | 72.2 | 51.6 | 58.9 | 45.2 | 53.5 | 63.2 |
| MCP ($\gamma = 0.7$) | **78.3** | **75.5** | **74.4** | **75.8** | **77.7** | 49.3 | 57.7 | 38.8 | 50.6 | 63.1 |
| MCP ($\gamma = 0.9$) | 58.3 | 52.3 | 56.1 | 54.2 | 56.6 | 46.2 | 50.8 | 38.7 | 46.6 | 50.5 |
| CPE ($\alpha = 0.01$) | 50.0 | 50.0 | 50.0 | 50.0 | 50.0 | 49.9 | 50.0 | 49.7 | 50.0 | 50.0 |
| CPE ($\alpha = 0.1$) | 51.4 | 50.4 | 51.1 | 51.0 | 51.4 | 47.2 | 49.9 | 43.7 | 47.7 | 49.1 |
| CPE ($\alpha = 0.3$) | 72.0 | 72.6 | 72.9 | 72.1 | 73.5 | 47.4 | 56.7 | 36.2 | 47.6 | 60.4 |
| DTC ($d = 10$) | 68.8 | 66.2 | 63.9 | 64.2 | 67.5 | 56.8 | 55.2 | 39.4 | 58.3 | 59.3 |
| DTC ($d = 20$) | 50.1 | 50.0 | 50.0 | 50.1 | 50.1 | 50.1 | 50.0 | 50.0 | 50.1 | 50.1 |
| DTC ($d = 30$) | 50.0 | 50.0 | 50.0 | 50.0 | 50.0 | 50.0 | 50.0 | 50.0 | 50.0 | 50.0 |
| ConfidNet (Corbière et al., 2019) | 73.4 | 69.0 | 67.0 | 68.6 | 71.1 | 58.8 | 59.8 | 46.8 | 60.7 | 63.2 |
| TrustScore (Jiang et al., 2018) | 74.4 | 67.5 | 64.8 | 67.6 | 70.8 | 57.9 | 59.7 | 48.1 | 60.5 | 62.6 |
| SSL (Luo et al., 2021) | 66.7 | 67.2 | 68.0 | 66.1 | 66.9 | 58.1 | 56.7 | 43.4 | 59.2 | 60.9 |
| SuperMentor (ours) | 70.9 | 65.5 | 61.7 | 67.5 | 68.8 | **81.6** | **64.1** | **68.5** | **81.9** | **70.0** |

# F. Relationship between mentor generalization and natural adversarial samples

As discussed in **Sec. 4.4**, we aim to examine the relationship between the generalization ability of mentors and natural adversarial samples. Following the definition in (Hendrycks et al., 2021b), natural adversarial samples are those that consistently lead to incorrect predictions across a wide range of AI models. In line with this definition, for each error source, we define $N_1$ as the set of samples misclassified by both ResNet50 and ViT mentees, serving as an analogue to the natural adversarial samples in (Hendrycks et al., 2021b). The samples misclassified only by the ResNet50 mentee (the ViT mentee can classify them correctly) are labeled set $N_2$, and those misclassified only by the ViT mentee (the ResNet50 mentee can

*Table S3.* **Role of natural adversarial samples in SuperMentor's generalization across mentee architectures in the IN dataset.** Each cell is formatted as: [number of samples correctly predicted by the mentor] / [total number of samples in the set] ([mentor prediction accuracy for this set]).

| Error Source | $N_1$ | $N_2$ | $N_3$ |
|:---:|:---:|:---:|:---:|
| ID | 1750/2225 (78.7%) | 1055/1352 (78.0%) | 429/561 (76.5%) |
| SpN | 2568/2973 (86.4%) | 2310/3075 (75.1%) | 462/579 (79.8%) |
| GaB | 2710/3078 (88.0%) | 1368/1657 (82.6%) | 642/757 (84.8%) |
| Spat | 2131/2587 (82.4%) | 1300/1627 (79.9%) | 464/584 (79.5%) |
| Sat | 2433/2880 (84.5%) | 1585/2012 (78.8%) | 502/618 (81.2%) |
| PGD | 3785/4689 (80.7%) | 377/709 (53.2%) | 2360/3286 (71.8%) |
| CW | 2528/3281 (77.0%) | 977/1952 (61.4%) | 1325/1886 (70.3%) |
| Jitter | 2527/3501 (72.2%) | 701/1032 (67.9%) | 1714/2685 (63.8%) |

classify them correctly) are labeled set $N_3$. The SuperMentor is trained on performance from the ResNet50 mentee and then generalizes its error prediction ability to the ViT mentee.

The result in **Appendix, Tab. S3** indicates that the natural adversarial samples in $N_1$ play a key role in explaining SuperMentor's strong generalization across different architectures. However, SuperMentor can also perform well on the non-natural adversarial samples ( $N_2$ and $N_3$ ). For example, it achieves 76.5% accuracy on the set $N_3$ for ID error types. This demonstrates that SuperMentor's robust generalization ability is not limited solely to natural adversarial samples.

## G. Visualization

To gain an intuitive understanding of SuperMentor's binary classification performance in error prediction, as mentioned in **Sec. 4.5**, we present the visualization of SuperMentor's embeddings on three types of error sources of a mentee in **Appendix, Fig. S3**. It is evident that SuperMentor can effectively segregate samples correctly classified by the mentee from those that are misclassified, forming two distinct clusters.

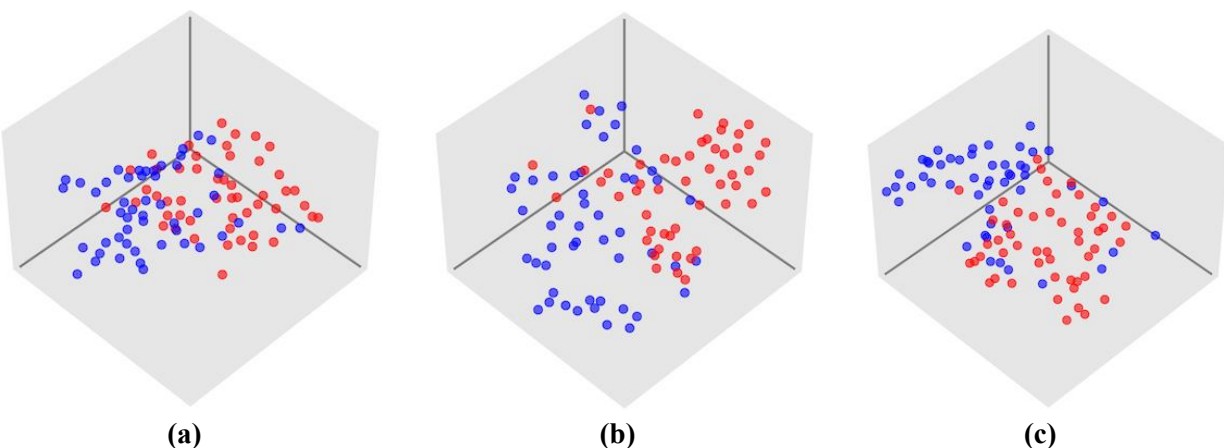

**(a)**        **(b)**        **(c)**

*Figure S3.* **3D visualization of the embeddings extracted from our SuperMentor Model for the classification of: a) C10-ID samples, b) C10-OOD-GaB samples and c) C10-AA-Jitter samples.** We use t-SNE (Van der Maaten & Hinton, 2008) to perform clusterings on the representations of our SuperMentor model for classifications of different error sources on the C-10 dataset. Red points indicate samples that the mentee fails to classify correctly, whereas blue points represent samples that the mentee successfully classifies. 50 red points and 50 blue points are randomly selected from the test sets and presented here. The visualized features are the embeddings in the second stream of the SuperMentor. Specifically, they are extracted before the final binary classification layer on whether the mentee makes a mistake.

## H. SuperMentor in real-world practice

To reflect the real-world contribution of our work, as mentioned in **Sec. 5**, we expanded our experiments to the NCTCRCHE100K (Kather et al., 2019) dataset which is used for medical image classification on colorectal cancer. This

dataset comprises around 100K images across 9 tissue classes. Predicting AI models' errors in medical image classification has significant real-world implications, such as reducing misdiagnoses and increasing the reliability of AI-assisted medical tools. **Appendix, Tab. S4** demonstrates that our SuperMentor accurately predicts the correctness of mentee outputs with an average accuracy of 81.8%, significantly outperforming other baseline methods. This highlights that our proposed framework can offer greater value and reliability for AI error prediction compared to existing approaches in medical image classification.

Given the increasing integration of AI models into our daily lives, ensuring their accuracy is paramount, especially in high-stakes fields such as medicine and finance. This experiment with medical image datasets underscores the critical role that SuperMentor plays in enhancing reliability and trustworthiness in these vital areas, showcasing its potential to make a meaningful impact where precision and dependability are essential.

*Table S4.* **Performance comparison of baselines (multiple hyperparameter configurations) and our SuperMentor on various error sources from the NCTCRCHE100K (Kather et al., 2019) dataset.** The ResNet50 model serves as the mentee, with its correctness of predictions evaluated by mentors. Best results are in bold.

| | ID | SpN | GaB | Spat | Sat | PGD | CW | Jitter | PIFGSM | Average |
|---|---|---|---|---|---|---|---|---|---|---|
| SER | 50.6 | 50.1 | 50.1 | 51.1 | 50.4 | 50.0 | 51.4 | 50.2 | 49.8 | 50.4 |
| MCP ($\gamma = 0.5$) | 53.0 | 59.1 | 53.0 | 53.2 | 50.6 | 53.8 | 53.3 | 53.2 | 54.8 | 53.8 |
| MCP ($\gamma = 0.7$) | 62.8 | 70.5 | 64.6 | 64.7 | 50.2 | 65.2 | 64.0 | 63.2 | 65.2 | 63.4 |
| MCP ($\gamma = 0.9$) | 72.5 | 78.5 | 72.1 | 75.0 | 48.8 | 77.0 | 73.8 | 76.0 | 75.2 | 72.1 |
| CPE ($\alpha = 0.01$) | 76.2 | 75.2 | 80.5 | 77.7 | 44.7 | 76.0 | 75.1 | 76.6 | 75.1 | 73.0 |
| CPE ($\alpha = 0.1$) | 75.5 | **79.5** | 76.4 | 76.2 | 48.2 | 77.8 | 76.2 | 77.6 | 76.9 | 73.8 |
| CPE ($\alpha = 0.3$) | 64.9 | 74.7 | 66.9 | 66.4 | 51.6 | 69.2 | 65.8 | 68.4 | 67.0 | 66.1 |
| DTC ($d = 10$) | 78.9 | 66.2 | 77.8 | 80.8 | **67.1** | 79.2 | 78.0 | 78.3 | 79.1 | 76.1 |
| DTC ($d = 20$) | 70.6 | 47.8 | 69.9 | 62.9 | 52.6 | 60.4 | 65.1 | 63.7 | 60.0 | 61.4 |
| DTC ($d = 30$) | 51.8 | 49.5 | 52.8 | 50.3 | 50.0 | 50.8 | 51.6 | 51.8 | 50.8 | 51.0 |
| ConfidNet (Corbière et al., 2019) | 85.1 | 61.3 | 86.2 | 74.0 | 58.4 | 79.3 | 82.9 | 81.8 | 79.4 | 76.5 |
| TrustScore (Jiang et al., 2018) | 73.3 | 60.0 | 73.4 | 62.2 | 62.4 | 71.0 | 72.8 | 71.4 | 71.5 | 68.6 |
| SSL (Luo et al., 2021) | 85.3 | 58.2 | 87.8 | 75.5 | 55.4 | 77.6 | 82.8 | 79.7 | 78.0 | 75.6 |
| SuperMentor (ours) | **88.5** | 64.1 | **89.2** | **84.1** | 56.9 | **88.2** | **88.7** | **88.3** | **88.3** | **81.8** |

## I. SuperMentor performance on additional OOD domains on ImageNet dataset

As mentioned in **Sec. 5**, we included experiments to evaluate the SuperMentor in more diversified OOD domains on ImageNet datasets, including ImageNet9-MIXED-RAND (IN9-MR) (Xiao et al., 2021), ImageNet9-MIXED-SAME (IN9-MS) (Xiao et al., 2021), ImageNet9-MIXED-NEXT (IN9-MN) (Xiao et al., 2021), ImageNet-R (IN-R) (Hendrycks et al., 2021a) and ImageNet-Sketch (IN-S) (Wang et al., 2019). Specifically, the MIXED-RAND, MIXED-SAME, and MIXED-NEXT datasets are derived from 9 classes in ImageNet and contain varying amounts of background and foreground signals. These datasets aim to demonstrate that models often classify objects based on background cues (often spurious features), rather than the objects themselves. Specifically, MIXED-SAME, MIXED-RAND, and MIXED-NEXT represent images with random backgrounds from the same class, random backgrounds from a random class, and random backgrounds from the next class, respectively. The ImageNet-R dataset comprises images featuring artistic representations of objects, such as cartoons, community-generated art, and graffiti renditions. The ImageNet-Sketch dataset consists of sketch-like images that match the ImageNet validation set in both categories and scale.

To achieve a more thorough evaluation of our SuperMentor (see **Sec. 4.5**) in OOD domains, we assess its performance on these additional OOD datasets. It is worth noting that the SuperMentor learns from the mentee's errors on adversarial ImageNet images generated by the PIFGSM attack only. In other words, the SuperMentor was NOT trained on any of these additional ImageNet-related OOD datasets. Yet, the SuperMentor achieves average accuracies of 70.9%, 71.0%, 69.6%, 62.2%, and 62.6% on the IN9-MR, IN9-MS, IN9-MN, IN-R, and IN-S datasets respectively, surpassing the 50% chance level. Despite that our Supermentor model is a simple ViT, it is still quite remarkable to achieve above-chance error prediction performance. We hope our work inspires researchers to explore more sophisticated AI mentors, incorporating advanced architectures or loss functions, to further enhance the ability to predict the correctness of another AI model's outputs.

## J. Detailed performance of mentors across various error sources

As mentioned in the caption of **Fig. 3**, the detailed results of mentors across various error sources for the C10, C100, IN datasets are shown in **Appendix, Fig. S4**, **Fig. S5** and **Fig. S6** respectively.

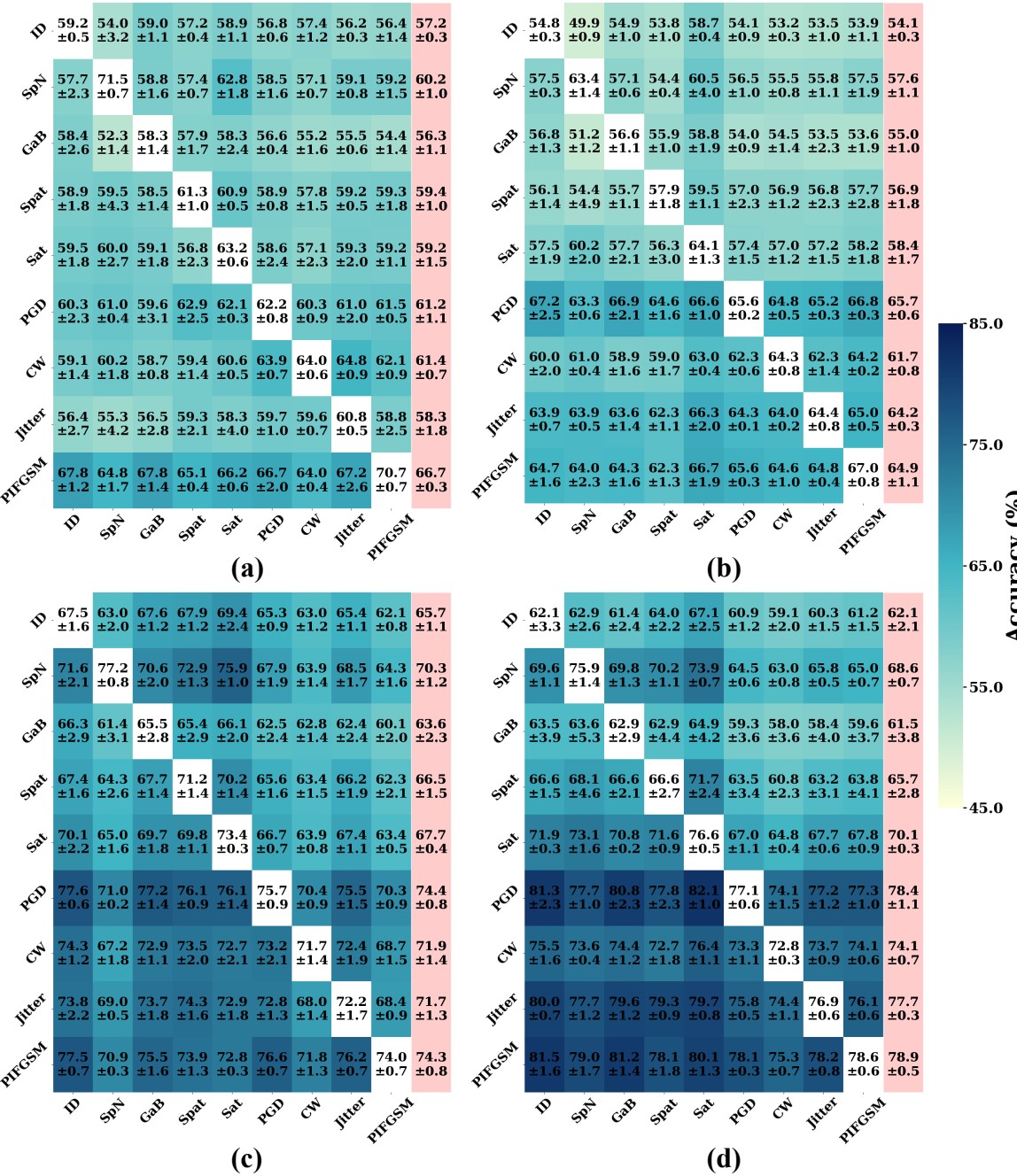

*Figure S4.* **Confusion matrices showing the average performance of mentor models across various error sources for the C10 dataset, presented in the format [mentee]-[mentor]: a) ResNet50-ResNet50, b) ViT-ResNet50, c) ResNet50-ViT, and d) ViT-ViT.** The confusion matrices' row labels indicate the training error source for the mentor, while the column labels denote the testing error sources for the mentor. Results in each cell denote the average accuracy with the standard deviation over 3 runs. The pink-highlighted column displays the row-wise mean and standard deviation over 3 runs.

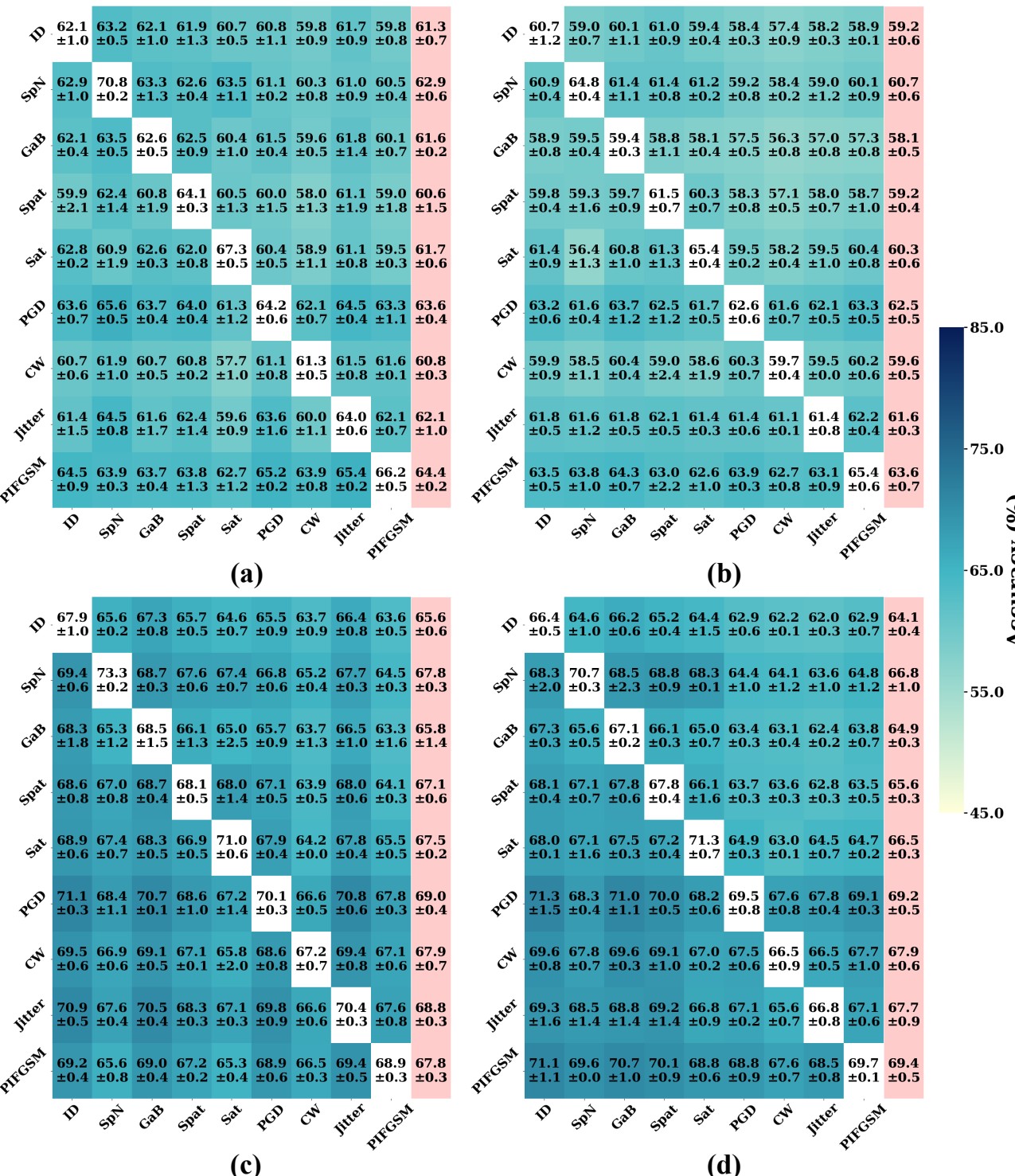

*Figure S5.* **Confusion matrices showing the average performance of mentor models across various error sources for the C100 dataset, presented in the format [mentee]-[mentor]: a) ResNet50-ResNet50, b) ViT-ResNet50, c) ResNet50-ViT, and d) ViT-ViT.** The confusion matrices' row labels indicate the training error source for the mentor, while the column labels denote the testing error sources for the mentor. Results in each cell denote the average accuracy with the standard deviation over 3 runs. The pink-highlighted column displays the row-wise mean and standard deviation over 3 runs.

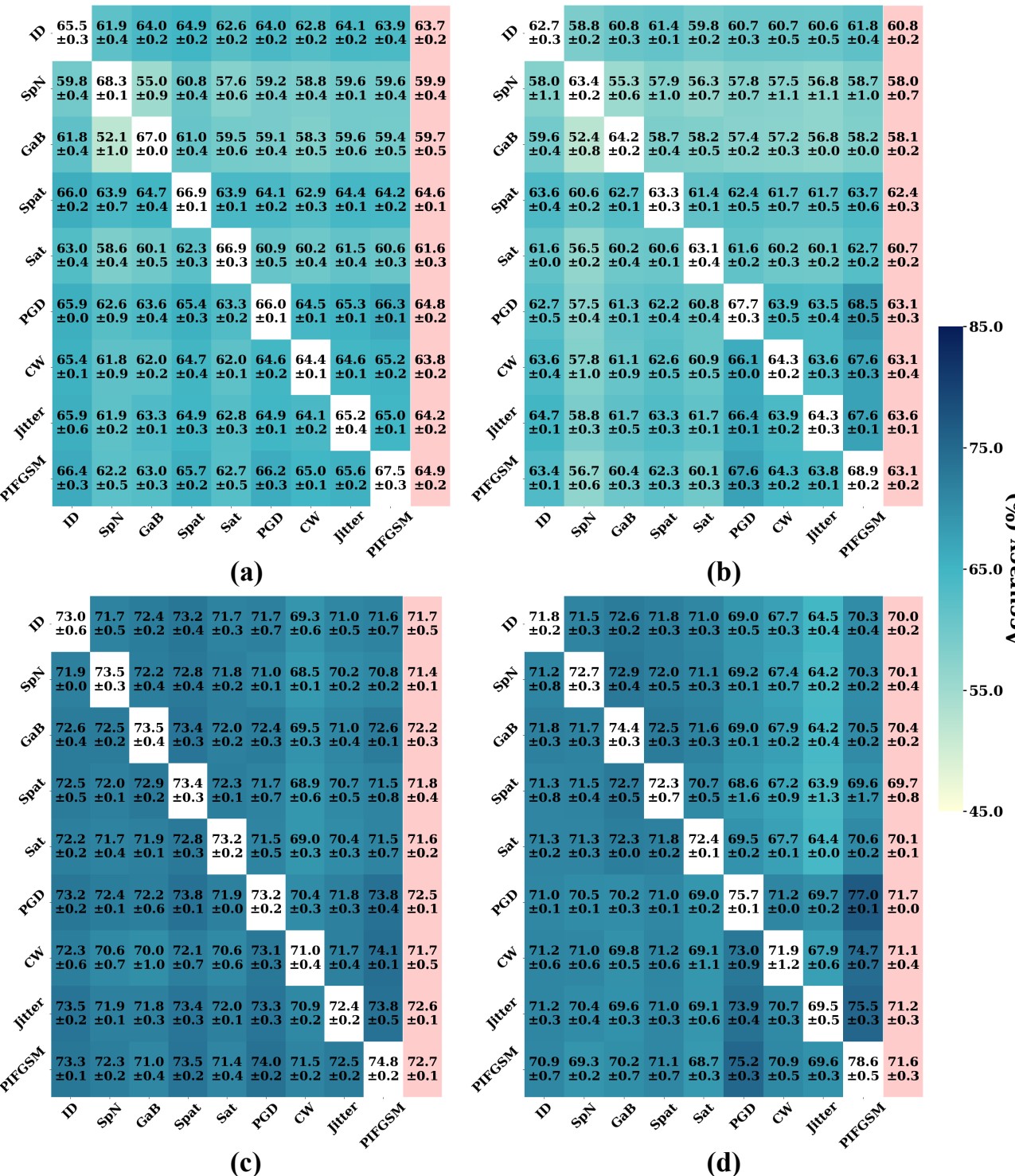

*Figure S6.* **Confusion matrices showing the average performance of mentor models across various error sources for the ImageNet-1K dataset, presented in the format [mentee]-[mentor]: a) ResNet50-ResNet50, b) ViT-ResNet50, c) ResNet50-ViT, and d) ViT-ViT.** The confusion matrices' row labels indicate the training error source for the mentor, while the column labels denote the testing error sources for the mentor. Results in each cell denote the average accuracy with the standard deviation over 3 runs. The pink-highlighted column displays the row-wise mean and standard deviation over 3 runs.

