# OpenReview forum: "Unveiling AI's Blind Spots: An Oracle for In-Domain, Out-of-Domain, and Adversarial Errors"
_ICML.cc/2025/Conference — ICML 2025 poster_

### Official Review · Reviewer_YWy5 · 2025-03-13

**Overall Recommendation:** 3

**Summary:**

The authors trained a secondary model (a mentor network) to predict whether a deep learning model would make a mistake or not. A mentor network consists of a backbone and two MLPs. The first MLP is trained to replicate latent variables of the mentee network whose answers are analyzed, and the second MLP is trained to predict the correctness of the mentee network. More specifically, the authors analyzed mentors’ accuracy on in-domain, out-of-domain and adversarial inputs.

**Claims And Evidence:**

The topic is of high importance because probing deep learning models’ decision-making process will help us build reliable models, but it remains unclear what we can learn from the authors’ reported results. First, the reported accuracy of mentor networks is not much higher than the baseline measures. Second, a mentor network is another black box, meaning we still do not fully understand their decision-making process. The authors speculated that mentor network errors can reveal the loss landscape of mentee networks (see loss landscape analyses), but they do not provide any quantitative results. If they can provide some evidence supporting that mentors can explain decision-boundary (or even loss landscape) to mentees, their study may have a profound impact.

**Essential References Not Discussed:**

N/A

**Experimental Designs Or Analyses:**

See my comments above.

**Methods And Evaluation Criteria:**

The authors compare their own algorithm with multiple baselines using multiple datasets, which is a strong point.

**Other Comments Or Suggestions:**

N/A

**Other Strengths And Weaknesses:**

N/A

**Questions For Authors:**

Q1: It is not clear why a mentor network would need two MLPs, each of which is independently trained. What happens if the first MLP, trained on the embedding of the mentee network, is removed?
Q2: The authors created out-of-domain examples by adding noises. As the corrupted inputs have the same semantic meanings, I do not think they are out-of-domain inputs. Can the authors further explain why the corrupted inputs are used as out-of-domain inputs?

Q3: The authors found that mentors can generalize across mentees. This result may be related to natural adversarial examples that many ImageNet models fail to correctly classify. Can the authors provide any insights into the implications of natural adversarial examples for the mentor-based analysis?

Reference for natural adversarial examples: Hendrycks, D., Zhao, K., Basart, S., Steinhardt, J., & Song, D.X. (2019). Natural Adversarial Examples. 2021 IEEE/CVF Conference on Computer Vision and Pattern Recognition (CVPR), 15257-15266.

**Relation To Broader Scientific Literature:**

The topic is highly important. But I do not think using another black-box model (i.e., mentor) can help us to better understand deep learning models (i.e., mentee) decision-making. The results presented are not sufficient.

**Theoretical Claims:**

There are no theoretical claims.

---

> ### Author Rebuttal · Authors · 2025-03-31
>
> **[YWy5.1-Claims And Evidence]**  First, our paper serves as a proof-of-concept, showing that training mentors with adversarial attack (AA) errors from the mentee has a greater impact on improving error prediction accuracy than training with in-domain (ID) or out-of-domain (OOD) errors. In **Tab. 1**, we use the adversarial images produced in the final iteration of the PIFGSM method to train SuperMentor.
>
> As requested by the reviewer, to probe the decision boundary of the mentees, we used adversarial images from all iterations of the PIFGSM method (with 3 iterations in total) to train our SuperMentor and named this new SuperMentor as "En-SuperMentor". By doing so, we can further diversify the samples and better capture the mentee's decision boundary. As shown in **Tab. R2 in [link](https://drive.google.com/file/d/1JtoYDxTmAcNfgN_6SVtoBGO1r7u49T6Q/view?usp=sharing)**, En-SuperMentor boosts average accuracy from 77.0% to 84.1%, which is significantly better than the other baselines and original SuperMentor.
>
> Next, as requested, we also added two quantitative loss landscape analyses using the method in [b]. In these analyses, we apply small perturbations to the mentor's weights and monitor the resulting changes in SuperMentor’s average accuracy.
>
> In the first loss landscape analysis (**Fig. R4 in [link](https://drive.google.com/file/d/1JtoYDxTmAcNfgN_6SVtoBGO1r7u49T6Q/view?usp=sharing)**), we examine mentors trained on three different error types: ID, OOD, and AA from the ImageNet-1K dataset. The results indicate that mentors trained on AA errors exhibit a wider loss landscape than those trained on ID or OOD errors. This suggests that training mentors with adversarial images makes them more robust to weight perturbations, thereby enhancing their ability to capture the generic features for predicting the mentee's decision-making. This finding strongly supports our loss landscape analysis discussed in **Sec. 4.1**.
>
> The second loss landscape analysis (**Fig. R5 in [link](https://drive.google.com/file/d/1JtoYDxTmAcNfgN_6SVtoBGO1r7u49T6Q/view?usp=sharing)**) examines the mentor’s error prediction performance using two different backbones: ResNet50 and ViT. The figure shows that mentors with ViT architectures have a considerably wider loss landscape compared to those with ResNet50. This further reinforces our claims in **Sec. 4.2** that transformer-based mentor models outperform their ResNet-based counterparts in error prediction.
>
> We will add these results and discussions in the final version.
>
> [b] Li et al. "Visualizing the loss landscape of neural nets." NeruIPS 2018.
>
> **[YWy5.2-Questions For Authors]** The MLP branch associated with distillation loss $L_d$ enables the mentor to mimic the mentee's predictions, and the MLP branch associated with loss $L_r$ ​is responsible for predicting whether the mentee will make mistakes on the input images. Without the first MLP for $L_d$​, the mentor cannot effectively capture the learning patterns of the mentee within its feature extraction backbone. We conducted an ablation study (see **Appendix, Tab. S4**) by removing the first MLP (i.e., eliminating the distillation loss $L_d$). The experimental results indicate that excluding the first MLP leads to a significant decrease in the mentor’s performance across all datasets.
>
> **[YWy5.3-Questions For Authors]** The concept of out-of-domain data remains somewhat ambiguous (Guérin, et al., 2023) in computer vision. In our paper, we define out-of-domain data as any data that falls outside the training domain (see **Sec. 3.3**), following the settings outlined in (Hendrycks & Gimpel, 2016; Luo et al., 2021; Yu et al., 2022). Although corrupted images have the same semantic meaning as clean images, the corruption methods introduce distributional shifts relative to the training domain. Whether these noisy images are classified as ID or OOD is irrelevant; what truly matters is that our SuperMentor can predict the errors of mentees when noise is added to the original images.
>
> **[YWy5.4-Questions For Authors]** According to the suggested paper by the reviewer, the natural adversarial samples refer to the samples that always lead to wrong predictions regardless of which AI model we use. Following the same definition, we defined sets $N_1, N_2, N_3$ in the caption of **Tab. R3 in [link](https://drive.google.com/file/d/1JtoYDxTmAcNfgN_6SVtoBGO1r7u49T6Q/view?usp=sharing)**. The result in **Tab. R3** indicates that the natural adversarial samples in $N_1$ play a key role in explaining SuperMentor’s strong generalization across different architectures. However, SuperMentor can also perform well on the non-natural adversarial samples ($N_2$ and $N_3$​). For example, it achieves 77.7% accuracy on the set $N_3$ for ID error types. This demonstrates that SuperMentor’s robust generalization ability is not limited solely to natural adversarial samples.
>
> We will cite the suggested paper and add these new results in the final version.

---

> > ### Comment · Reviewer_YWy5 · 2025-04-08
> >
> > I appreciate the fact that the authors conducted more experiments, which further supports that mentor networks can learn mentee networks’ behaviors. Although I still am not sure what we can learn from the black-box models that are trained to explain other black-box models, the authors’ empirical studies may be helpful to future experiments. Thus, I am raising my rating to 3.

---

> > > ### Author Response · Authors · 2025-04-09
> > >
> > > We sincerely appreciate the time and effort you have put into thoughtfully reviewing our paper. Your feedback has been helpful in enhancing our work.

---

### Official Review · Reviewer_B54K · 2025-03-14

**Overall Recommendation:** 3

**Summary:**

This paper proposes to use mentor models to predict the errors of mentee models. The mentor model can learn from the mistakes of the mentees on adversarial images and generalize to predict the in-domain and out-of-domain errors of the mentees. In addition, the mentor trained on a mentee generalizes well in predicting the errors of other mentees.

## update after rebuttal
The rebuttal solved my concerns, and I will keep my positive rating after the rebuttal if the authors could incorporate the clarification into their draft.

**Claims And Evidence:**

The claims are supported by experimental evidence.

**Essential References Not Discussed:**

I didn't identify any such references.

**Experimental Designs Or Analyses:**

The experiment design makes sense in this problem setting, and the authors compared the performance of their proposed method with a series of baselines.

**Methods And Evaluation Criteria:**

The proposed methods and evaluation make sense for the problem setting.

**Other Comments Or Suggestions:**

NaN

**Other Strengths And Weaknesses:**

Strengths:

This paper proposes to predict model errors in a unified way, including errors on the in-domain, out-of-domain, and adversarial images. The mentor model is also generalizable to other mentee models and error types. This paper is well-written and easy to follow.

Weaknesses:

This work focuses on the accuracy of predicting various errors for mentee models but does not distinguish the specific error type. In addition,  applying mentor models requires training and aligning with the mentee models, which relies on the accessibility of mentee models and training data and, therefore, might limit its applicability. In addition, the evaluation of some common shifts between training and test data might be worth exploring, such as errors from spurious correlation, which can be categorized as ID errors and have great potential in applications.

**Questions For Authors:**

Except for the weaknesses I mentioned above, I only have minor questions:

1. Given that mentor models can recognize ID errors, will the mentor model facilitate the training of mentee models?

2. The paper mentioned that the joint training with ID, OOD, and AA data marginally outperforms the mentor trained only with AA. Was the number of samples the same for joint training or AA training? The improvement might be from the increased number of samples

**Relation To Broader Scientific Literature:**

Previous relevant work on error monitoring relies on the manually defined metrics to estimate the likelihood of errors, while this work aims to train a mentor model to predict various errors that might happen in mentee model predictions.

**Theoretical Claims:**

There is no theoretical claim in this paper.

---

> ### Author Rebuttal · Authors · 2025-03-31
>
> **[B54K.1-Other Strengths And Weaknesses]** Thank you for your insightful comments! We address the points raised as follows:
>
> First, training a mentor to distinguish specific error types is a promising future research direction, as we have mentioned in **Sec. 5** of our paper. This task is more challenging than predicting whether a mentee will make an error. It requires a mentor to first achieve high accuracy in error prediction before extending its capabilities to distinguish the exact types of errors a mentee may encounter. Therefore, our work serves as a solid foundation for this promising research direction.
>
> Second, we agree that our method requires access to the mentee's model parameters to generate adversarial images. Training the mentor without using these parameters (i.e., treating the mentee as a black box) presents an interesting and challenging future research direction. However, we respectfully disagree that our mentor needs access to the mentee's training data. Our mentor is instead trained solely on the mentee's behaviors on its evaluation datasets. This is a significant advantage for real-world applications, especially when training the mentee is resource-intensive (e.g., high memory usage) or when the training data of a mentee is inaccessible.
>
> Third, as described in **Appendix, Sec. I**, we conducted experiments evaluating our SuperMentor on additional OOD domains of the ImageNet dataset, including errors arising from spurious correlations. Specifically, we tested our SuperMentor on datasets such as ImageNet9-MIXED-RAND [a], ImageNet9-MIXED-SAME [a], and ImageNet9-MIXED-NEXT [a], which include images with random backgrounds from a random class, random backgrounds from the same class, and random backgrounds from the subsequent class, respectively. These datasets are designed to reveal how models can misclassify objects by focusing on spurious background cues rather than the objects themselves. For instance, an AI model might label an object as a bird based solely on a tree in the background. The experimental results demonstrate that our SuperMentor can still achieve above-chance error prediction performance on these datasets.
>
> We will include these discussions in the final version of our paper.
>
> [a] Xiao, et al. "Noise or Signal: The Role of Image Backgrounds in Object Recognition." ICLR 2021.
>
> **[B54K.2-Questions For Authors]** Thank you for the reviewer's insightful question! As mentioned in **Sec. 5** of our paper, the reviewer’s point is indeed a promising future research direction. We would like to emphasize that our work serves as an important foundation in this research direction, as mentors with higher error prediction accuracy may enhance the training of mentee models by correcting their mistakes on the fly.
>
> **[B54K.3-Questions For Authors]** We would like to clarify that the number of samples differs between Joint training and AA training. According to **Appendix, Tab. S1**, Joint training uses 25092 samples, whereas AA training uses only 6774 samples. The marginal improvement observed in Joint training may be attributed to the larger training sample size. This suggests that the quantity of samples is not the key factor in enhancing a mentor’s error prediction performance. Training mentors on samples that accurately capture the error patterns of mentees is crucial. For instance, the adversarial images generated using the PIFGSM method in our paper effectively illustrate this point. With only 6774 training samples from PIGFSM—one-fourth the size of the Joint training dataset—our SuperMentor on AA training achieves performance comparable to Joint training. These findings further underscore the significance of our study by providing critical insights into optimal training practices for mentor models.

---

> > ### Comment · Reviewer_B54K · 2025-04-04
> >
> > Thanks for the rebuttal from the authors and I will keep my rating. I agree that it would be an advantage if the mentor model only needs the evaluation datasets and the discussion on the comparison of joint and AA training makes sense to me. In addition, does the joint training use all samples from ID, OOD, and AA in Tab S1? A minor confusion is that I'm not sure which datasets and models correspond to "Joint training uses 25092 samples, whereas AA training uses only 6774 samples" in the rebuttal. The ResNet mentee with CIFAR 10 described in Appendix D has a total number of 5766 test samples for AA and less than 10k test samples for the sum of ID, OOD, and AA.

---

> > > ### Author Response · Authors · 2025-04-06
> > >
> > > For Joint training, we do NOT use all the samples from ID, OOD and AA listed in **Appendix, Tab. S1**. Instead, as described in **Appendix, Sec. D**, we select samples from the speckle noise (SpN) error source for the OOD error type and from the PIFGSM error source for the AA error type, along with ID samples, to train the mentor. We select SpN and PIFGSM because these two error sources are most effective for training mentors in their respective error types. In other words, mentors trained on SpN achieved the highest average accuracy among the OOD-trained mentors, while those trained on PIFGSM attained the best average accuracy among the AA-trained mentors (see **Appendix, Fig. S3(c)**).
> > >
> > > Therefore, based on the setup described above, in Joint training, ResNet50 serves as the mentee, and the training samples are drawn from the CIFAR-10 dataset’s ID, SpN, and PIFGSM error sources. The total number of training samples of Joint training is calculated as 151 + 9547 + 690 + 7930 + 1613 + 5161 = 25092. For AA training, only the training samples from the PIFGSM error source are used, resulting in 1613 + 5161 = 6774 training samples. Please refer to **Appendix, Tab. S1** for these numbers.
> > >
> > > In addition, during this rebuttal, we also introduced a fully joint trained mentor which was trained on all error sources (ID, OOD, and AA, a total of 9 error sources), achieving an average evaluation accuracy of 84.6% across all error sources. In this scenario, all testing samples are in distribution because the mentor has seen samples from every error source during training. Despite this, the improvement compared to a mentor trained solely on the PIFGSM error source (84.6% vs. 78.0%) is insignificant. Consequently, this finding further indicates that (1) Training mentors exclusively on the AA error source offers a more data-efficient strategy for real-world applications. (2) Remarkably, a SuperMentor trained solely on AA generalizes well to unseen error sources, achieving performance comparable to the upper bound obtained by jointly training on all error sources.

---

### Official Review · Reviewer_LEYc · 2025-03-21

**Overall Recommendation:** 3

**Summary:**

The authors propose training a "mentor" model to learn to predict the errors of a "mentee" model. The authors evaluate multiple choices for training the mentor model, such as training with in-distribution, out-of-distribution, and adversarial examples to predict errors. The authors combine these results to propose a "SuperMentor" which can outperform the baseline mentors in predicting errors.

**Claims And Evidence:**

Strengths:
- The paper trains and evaluates their mentor models on a number of settings, across different architectures and datasets, and compares to many relevant baselines.
- The discussion regarding what training data (between in-distribution, out-of-distribution, and adversarial examples, plus some investigation of level of distortion of the images) is very interesting
- Investigating the role of how different architectures affect the the accuracy was also interesting.
- The paper tackles an important problem of predicting when neural networks make errors.


Weaknesses:
- The main technical weakness is the evaluation. In particular, the authors choose to select a data split for each model and each dataset where half of the examples were classified correctly and half of the examples were classified incorrectly. This means that each model is evaluated on different datasets. Plus, after generating the dataset, the remaining examples were used for training, and so the training differ from model to model and setting to setting. This makes comparisons across settings very hard to evaluate. For example, there is a confounding variable where it is unclear if adversarial vs OOD vs ID examples carry more information about predictions themselves, or if the examples for which the models make errors are easier/harder to learn. I think to address this the authors should consider how to control for this setting—or, if a reasonable method to control for this doesn't exist—at least ablating the choice of exactly which examples were chosen for training and evaluation. While this is hopefully easily addressable, I believe that this issue can potentially confound the takeaways from the paper, and so it should be addressed before publication. If this were addressed at all my score would increase.
- A more minor issue: In, e.g., Figure 3, the variance appears quite large. It would be likely important to run some sort of a statistical test to ensure that the differences are in fact statistically significant before making any claims. Further, Table 1 should include some measure of variance to understand if the difference in performance between the proposed method and baselines (which is itself relatively small) is a result of noise.

**Essential References Not Discussed:**

No essential references are not discussed, as far as I can tell.

**Experimental Designs Or Analyses:**

Please refer to Claims and Evidence for comments.

**Methods And Evaluation Criteria:**

The methods and evaluation criteria make sense, although please refer to Claims and Evidence for additional comments on the validity of the methods.

**Other Comments Or Suggestions:**

N/A

**Other Strengths And Weaknesses:**

See above.

**Questions For Authors:**

N/A

**Relation To Broader Scientific Literature:**

The contributions are connected to relevant literature.

**Theoretical Claims:**

N/A

---

> ### Author Rebuttal · Authors · 2025-03-31
>
> **[LEYc.1-Claims And Evidence]** We appreciate the reviewer’s thoughtful question and would like to clarify the following two points:
>
> First, our evaluation settings are fair for all mentors. The dataset split presented in **Appendix, Tab. S1** applies only to the error source used for training the mentor, not to all error sources. For instance, if Mentor A is trained on adversarial images generated by PIFGSM error source, only the PIFGSM-generated samples are split into training and evaluation sets according to **Appendix, Tab. S1**. All samples from other ID, OOD, and AA error sources are fully used for evaluation. Thus, comparing its performance with that of other mentors on these error sources is fair. It is true that Mentor A may perform better on the PIFGSM error source since the mentor was trained on it. However, even when excluding its performance on the PIFGSM error source, its overall average performance ranking compared to other mentors remains largely unchanged. This observation applies to mentors trained on other error sources as well. Confusion matrices in **Appendix, Fig. S3, S4, and S5** show that the evaluation results on the trained error sources (accuracy in the white background of confusion matrices) do not dominate the average performance (accuracy in the pink background of confusion matrices).
>
> Second, we appreciate the reviewer's question regarding a potential confounding variable in mentor training—specifically, whether AA, OOD, or ID examples inherently provide more predictive information, or if some error examples are simply easier or harder to learn. To address this, we adopted an alternative dataset splitting strategy for mentor training, distinct from the one in **Appendix, Tab. S1**. For example, we start with the original images from CIFAR-10 and separately apply two perturbation methods, PIFGSM and SpN, to produce two sets of error sources. Next, use 60% of the PIFGSM-generated images to train a mentor model, and reserve the remaining 40% for testing. Similarly, use 60% of the SpN-generated images (corresponding to the same original CIFAR-10 images as the PIFGSM training set) to train another mentor model, and reserve the remaining 40% (corresponding to the same original CIFAR-10 images as the PIFGSM testing set) for testing.
>
> This approach is applied consistently across mentors trained on other error sources, ensuring that all mentors are trained and evaluated on images derived from the same original CIFAR-10 images but different domain shifts. Consequently, the confounding factor related to the inherent ease or difficulty of learning certain samples is eliminated, leaving only the question of whether AA, OOD, or ID samples inherently provide more predictive information.
>
> **Figures R1, R2, and R3 in [link](https://drive.google.com/file/d/1JtoYDxTmAcNfgN_6SVtoBGO1r7u49T6Q/view?usp=sharing)** present the performance of mentors on various error sources for the CIFAR-10, CIFAR-100, and ImageNet-1K datasets, respectively, under this new data split strategy. The results reveal that training mentors with adversarial attack errors from the mentee leads to a greater improvement in error prediction accuracy compared to training with ID or OOD errors. Consistent with our original finding reported in the paper, this finding confirms that adversarial images carry more predictive information than either ID or OOD images.
>
> We will add these discussions and new experimental results in the final version.
>
> **[LEYc.2-Claims And Evidence]** Thank you for the suggestions! To support our claims in **Fig. 3**, we computed p-values to compare the error prediction accuracy of mentors trained with different error types. Our primary claim is that mentors trained on the adversarial attack (AA) error type achieve higher accuracy than those trained on in-domain (ID) or out-of-domain (OOD) error types. To validate this, we calculated p-values using a two-tailed t-test and compared the performance of AA-trained mentors to ID-trained mentors and AA-trained mentors to OOD-trained mentors across four different [mentee]-[mentor] settings (ResNet50-ResNet50, ViT-ResNet50, ResNet50-ViT, ViT-ViT) on three datasets: CIFAR-10, CIFAR-100, and ImageNet-1K. All pairwise p-values are below 0.05, indicating that these performance differences are statistically significant. Therefore, our claim is well-supported. We will include these p-value analyses in the final version.
>
> We also agree with the reviewer's suggestion regarding **Tab 1**. The updated version, now including variance over 3 runs, is shown in **Tab. R1 in [link](https://drive.google.com/file/d/1JtoYDxTmAcNfgN_6SVtoBGO1r7u49T6Q/view?usp=sharing)**. This confirms that the performance differences between our proposed method and the baselines are not simply attributable to noise. We will include this updated table in the final version.

---

> > ### Comment · Reviewer_LEYc · 2025-04-03
> >
> > Just to clarify—are all mentor models, regardless of their training data, evaluated on the exact same evaluation examples?
> >
> > On line 241 right column: "To create balanced training and test sets for the mentor and avoid the long-tailed distribution problem, we select an equal number of correctly and incorrectly classified samples for each training batch and every test set."
> >
> > This seems to imply that different mentors evaluate on different test sets, since different mentees (between ViT and ResNet50) get different examples correct and incorrect.

---

> > > ### Author Response · Authors · 2025-04-06
> > >
> > > TLDR:
> > > The reviewer rightly notes that the evaluation sets for the mentors in our original submission were not identical, although they were drawn from the same underlying distributions. In our rebuttal, we introduce more rigorous controls to address this issue and ensure that all mentors are evaluated using the exact same test set. Importantly, the results from these updated experiments do **not** change the overall conclusions of the paper. We will include these revised experiments in the updated version.
> > >
> > > Below, we provide detailed clarifications to directly address the reviewer’s concern.
> > >
> > > 1. The process described in line 241 (right column) applies exclusively to the training error source. For instance, if a mentor is trained using adversarial images generated by the PIFGSM error source, we ensure that both its training batch and the corresponding PIFGSM test set contain an equal number of correctly and incorrectly classified samples. In contrast, when evaluating this mentor on other error sources (like SpN or GaB), all available samples from other error sources are used. This approach is applied consistently across all mentors.
> > >
> > > 2. When comparing two mentors trained on the performance of **same** mentee using different error sources, each mentor is evaluated with **exact same** sample sets for every error source except the one used during training. For example, if the mentee is ResNet50 and we compare a mentor trained on the PIFGSM error source with one trained on the SpN error source, then the evaluation samples are **identical** across all error sources except for the PIFGSM and SpN error sources. The evaluation samples in PIFGSM and SpN error sources are determined using the data splitting strategy outlined in **Appendix, Tab. S1**. Importantly, even if we exclude their performance on the PIFGSM and SpN error sources, their overall average performance ranking compared to other mentors remains largely unchanged.  This observation applies to mentors trained on other error sources as well. Confusion matrices in **Appendix, Fig. S3, S4, and S5** show that the evaluation results on the trained error sources (accuracy in the white background of confusion matrices) do not dominate the average performance (accuracy in the pink background of confusion matrices). This addresses the concern that in-domain performance might overshadow the out-of-domain performance.
> > >
> > > 3. After realizing that the point 1 above might lead to unfair comparisons among all the mentors due to their different test sets, we introduced an alternative evaluation strategy in **[link](https://drive.google.com/file/d/1OzBkXlU1ROtJD66r7Zz1ISFZq69LeQeh/view?usp=sharing)** by assessing all mentors on the **exact same** balanced testing set—each error source's test set contains an equal number of samples that the mentee correctly and incorrectly classified. In this approach, mentors trained on the performance of the same mentee are evaluated on **identical** testing sets across all error sources. **Figures R1, R2, and R3 in [link](https://drive.google.com/file/d/1OzBkXlU1ROtJD66r7Zz1ISFZq69LeQeh/view?usp=sharing)** illustrate the mentors' performance across various error sources for the CIFAR-10, CIFAR-100, and ImageNet-1K datasets, respectively, under this new evaluation strategy. The results show that training mentors with adversarial attack errors from the mentee leads to a significantly greater improvement in error prediction accuracy compared to training with ID or OOD errors, which is consistent with our original findings.

---

### Decision · Program_Chairs · 2025-05-01

**Decision:**

Accept (poster)

**Comment:**

The paper conducts comprehensive empirical evaluations that train a “mentor” model to predict the behaviors of a “mentee” model. The topic is considered important as probing deep learning models’ decision-making processes will help us build reliable models.